# Can land degradation drive differences in the C exchange of two similar semiarid ecosystems?

Ana López-Ballesteros[1,2], Cecilio Oyonarte[3,4], Andrew S. Kowalski[5,6], Penélope Serrano-Ortiz[1,6], Enrique P. Sánchez-Cañete[5,6,7], M. Rosario Moya[2], Francisco Domingo[2]

[1]Departamento de Ecología, Universidad de Granada, Granada, 18071, Spain
[2]Departamento de Desertificación y Geo-ecología, Estación Experimental de Zonas Áridas (CSIC), Almería, 04120, Spain
[3]Departamento de Agronomía, Universidad de Almería, Almería, 04120, Spain
[4]Centro Andaluz de Evaluación y Seguimiento del Cambio Global (CAESCG). Universidad de Almería. Almería, 04120, Spain
[5]Departamento de Física Aplicada, Universidad de Granada, Granada, 18071, Spain
[6]Instituto Interuniversitario de Investigación del Sistema Tierra en Andalucía (IISTA-CEAMA), Universidad de Granada, Granada, 18006, Spain
[7]B2 Earthscience, Biosphere 2, University of Arizona, Tucson, Arizona, 85721, USA

*Correspondence to*: Ana López-Ballesteros (alpzballesteros@gmail.com)

**Abstract.** Currently, drylands occupy more than one third of the global terrestrial surface and are recognized as areas vulnerable to land degradation. The concept of land degradation stems from the loss of an ecosystem's biological productivity, due to long-term loss of natural vegetation or depletion of soil nutrients. Drylands´ key role in the global carbon (C) balance has been recently demonstrated, but the effects of land degradation on C sequestration by these ecosystems still need to be investigated. In the present study, we compare net C and water vapour fluxes, together with satellite, meteorological and vadose zone ($CO_2$, water content and temperature) measurements, between two nearby (~23 km) experimental sites representing "natural" (i.e. site of reference) and "degraded" grazed semiarid grasslands. We utilized data acquired from two eddy covariance stations located in SE Spain during 6 years with highly variable precipitation magnitude and distribution. Results show a striking difference in the annual C balances with an average release of $196 \pm 40$ and $-23 \pm 20$ g C $m^{-2}$ $yr^{-1}$ for the "degraded" and "natural" sites, respectively. At the seasonal scale, differing patterns in net $CO_2$ fluxes were detected over both growing and dry seasons. As expected, during the growing seasons, greater net C uptake over longer periods was observed at the "natural" site, however, much greater net C release, probably derived from subterranean ventilation, was measured at the "degraded" site during drought periods. After subtracting the non-biological $CO_2$ flux from net $CO_2$ exchange, flux partitioning results point out that, during the six years of study, gross primary production, ecosystem respiration and water use efficiency were, on average, nine, twice and ten times higher, respectively, at the "natural" site *versus* the "degraded" site. We also tested differences in all monitored meteorological and soil variables and found it most relevant that $CO_2$ at 1.50 m belowground was around 1000 ppm higher in the "degraded" site. Thus, we believe that subterranean ventilation of this vadose zone $CO_2$, previously observed at both sites, largely drives the differences in C dynamics between them, especially during the dry season maybe due to enhanced subsoil-atmosphere interconnectivity

in the "degraded" site. Overall, the 12 site-years of data allow direct exploration of the roles of climate and land degradation in the biological and non-biological processes that ultimately control the C sequestration capacity of semiarid ecosystems.

## 1 Introduction

The concept of land degradation stems from the loss of an ecosystem's biological productivity, which in turn relies on several degradation processes such as long-term loss of natural vegetation, deterioration of soil quality, biodiversity depletion or water and wind erosion (UNCCD, 1994). Drylands (arid, semiarid and dry sub-humid areas), which occupy more than one third of Earth´s land surface and are inhabited by more than 2 billion people (Niemeijer et al., 2005),  have been recognized as areas vulnerable to land degradation processes.  In fact, they have expanded globally for the last sixty years at an estimated annual rate of 5.8 million hectares in mid latitudes alone (Lal, 2001), and are projected to expand under future climate change scenarios (Feng and Fu, 2013; Cook et al., 2014), especially in the Mediterranean region, where major expansions of semiarid areas will occur (Gao and Giorgi, 2008; Feng and Fu, 2013).

Over recent decades, most research focused on land degradation has been based on remote sensing and earth observation techniques. Much of these investigations have aimed to refine methodological issues in order to accurately track land degradation in vulnerable areas, reduce uncertainties and explain inconsistencies among studies. For instance, a wide array of satellite-derived data, such as vegetation indices, normalized surface reflectance, brightness temperature or biomass-net primary production derivatives (Mbow et al., 2015),  has been utilized to appraise desertification effects in the Sahel (Mbow et al., 2015; Fensholt et al., 2013) and also in other African countries such as Kenya (Omuto, 2011), Somalia (Omuto et al., 2010), South Africa (Thompson et al., 2009) or Zimbabwe (Prince et al., 2009). Likewise, desertification in the Mediterranean region has been studied through satellite imagery in Greece (Bajocco et al., 2012), Israel (Shoshany and Karnibad, 2015) and the Iberian Peninsula (del Barrio et al., 2010). However, although drylands´ key role in the global carbon (C) balance has been demonstrated (Poulter et al., 2014; Ahlström et al., 2015), very few investigations have directly quantified how land degradation processes disturb the C sequestration capacity of drylands (Lal, 2001), despite being one of the most important ecosystem services (Watanabe and Ortega, 2011).

In this regard, the few C-related desertification studies conducted over last decade have centred on soil C dynamics. Concretely, soil organic carbon (SOC) inventories have been used to explore the effects of climate, human activities and grazing pressure in desertification-prone areas of China (Feng et al., 2006) and Brazil (Schulz et al., 2016). Similarly, other investigations have evaluated soil degradation processes by means of soil $CO_2$ effluxes together with other biometric measurements in drylands found in China (Hou et al., 2014; Wang et al., 2007), Chile (Bown et al., 2014) and southeast Spain (Rey et al., 2011;Rey et al., 2017). However, the degradation processes associated with desertification affect several subsystems as well as their interactions at multiple spatial and temporal scales. For instance, adverse effects on soil quality

involve depletion of soil fertility, but also reduce soil-water storage (Mainguet and Da Silva, 1998), which, in turn, can constrain seed germination and vegetation reestablishment, modify climax vegetation, disrupt biogeochemical cycles, alter water and energy balances, and consequently lead to a loss of ecosystem resilience (Lal, 2001). This cascade of disturbances may result in a reduction of the C sequestration capacity of a given ecosystem, which is clearly a symptom of the loss of

biological productivity, resulting in a positive feedback to global warming. Therefore, a quite suitable and holistic approach is to integrate all subsystems effects into a whole ecosystem-scale assessment when quantifying the C loss derived from land degradation. However, the use of this integrative method is mostly lacking in the available literature.

The present study is located in an area, the southeast of Spain, that has been recognized as a hotspot of land degradation

owing to the synergistic interaction of sociological and climatic factors (Puigdefábregas and Mendizabal, 1998). Our core aim is to evaluate how dryland degradation affects the dynamics of net ecosystem-atmosphere C exchange of two semiarid grasslands that represent differing degradation status ("natural" *versus* "degraded") by means of meteorological, satellite and subsoil $CO_2$ measurements together with carbon and water fluxes acquired by the eddy covariance (EC) technique (Baldocchi et al., 1988). Owing to the high temporal resolution of the EC method, we can assess the effect of land

degradation as a slow change or disturbance legacy in the studied ecosystems and how, in turn, it influences the ecosystems' resilience against short-term disturbances, such as climate extremes (i.e. droughts, heat waves).

Some land degradation processes are evident when we compare the "natural" site with the "degraded" site. Accordingly, our main hypothesis is that land degradation processes can directly affect abiotic and/or biotic factors and, consequently, influence the biological and/or non-biological processes that compose the net ecosystem $CO_2$ exchange: gross primary

production, ecosystem respiration (biological processes) and subterranean ventilation – a non-biological process that provokes the transfer of $CO_2$-rich air from subsoil to atmosphere under drought and high turbulence conditions. Firstly, the lower vegetation cover in the "degraded" site would entail a higher thermal and radiative stress at the surface, especially during the drought period (Rey et al., 2017). The hypothesized effects on biological processes are a direct reduction in plant productivity and respiration, and an indirect decrease in heterotrophic respiration. Secondly, the higher cover of bare soil and

outcrops in the "degraded" site may increase the soil-atmosphere interconnectivity, which indirectly can enhance the presence of advective $CO_2$ release through subterranean ventilation, which has been previously measured at both experimental sites (Rey et al., 2012; López-Ballesteros et al., 2017). And thirdly, the reduced soil fertility and depth may provoke changes in microbial communities (Evans and Wallenstein, 2014) due to stronger nutrient and water limitations. Consequently, a direct decrease in heterotrophic respiration and plant productivity and respiration is expected.

Hence, our specific objectives are: (1) to compare the C sequestration capacity of two semiarid ecosystems with differing degradation status, (2) to study the underlying processes (biological *versus* non-biological) and influencing factors that can drive potential differences in the net C exchange of studied ecosystems, and (3) to evaluate whether degradation can modulate ecosystem responses against short-term disturbances. To do this, we analysed 12 site-years of EC data, Enhanced

Vegetation Index (EVI) time series and monitored ambient variables registered over the same period (2009-2015) at both sites. Additionally, we used subsoil $CO_2$, moisture and temperature data obtained during 2014-2015.

## 2 Material and Methods

### 2.1 Experimental sites description

The study area is located in southeast Spain, the driest part of Europe. The two experimental sites, Amoladeras (N36º50'5''W2º15'1'') and Balsa Blanca (N36º56'26.0'' W2°01'58.8''), are found within the Cabo de Gata-Níjar Natural Park (Almería, Spain; Fig. 1) and are quite similar in terms of climate and ecosystem type. Both sites show a desert climate, according to Köppen classification (Bwh; Kottek et al., 2006), with a mean annual temperature of 18°C, and mean annual precipitation of approximately 220 mm.

The ecosystem type corresponds to *espartal*, a Mediterranean semiarid grassland where the dominant species is *Machrocloa tenacissima*. This ecosystem type is widely extended over the Western Mediterranean region;  in Cabo de Gata-Níjar Natural Park, a great fraction of agricultural areas that were abandoned over 1957-1994 resulted in *espartal* ecosystems (Alados et al., 2011; Alados et al., 2004). The functioning of both experimental sites can be divided into two main periods. On one hand, the growing season usually extends from late autumn to early spring, when the temperature starts to rise and water

resources have not yet become scarce (López-Ballesteros et al., 2016; Serrano-Ortiz et al., 2014). On the other hand, a long period of hydric stress, with high temperatures and scarce precipitation, results in a prolonged dry season that usually begins in May-June and ends in September-October, when the first autumn rainfall events occur. Additionally, water inputs derived from relevant dewfall episodes, which have been previously reported in the area (Uclés et al., 2014), can rehydrate soil and plants during night and early morning hours.

Regarding the topographic, geologic and edaphic characteristics, both sites are located on an alluvial fan, where the main geological materials consist of quaternary conglomerates and Neogene-Quaternary sediments cemented by lime (caliche) (Rodríguez-Fernández et al., 2015) on slopes of 2-6% (Rey et al., 2017) so no significant runoff occurs. Additionally, both sites present petrocalcic horizons. However, altitude and soil type differ. While Balsa Blanca (hereinafter BB) is located at an altitude of 208 m and has MollicLithic Leptosols (Calcaric), Amoladeras (hereinafter AMO) is situated closer to sea level,

at 65 m, and presents less developed soils Lithic Leptosol (Calcaric; Table 1).

Overall, as stated by Rey et al. (2011), these two experimental sites represent different degradation stages owing to their differing soil characteristics and surface fractions (Table 1). While BB has more deep and fertile soils and higher vegetation cover, AMO shows thinner and poorer soils and has half of Balsa Blanca´s vegetation cover. Therefore, in accordance with Rey et al. (2011; 2017), we considered that BB represents the "natural" site, being, currently, a representative ecosystem of 

the area, while AMO represents a "degraded" site with respect to BB. The stronger degradation effects observed in AMO ("degraded" site) compared to BB ("natural" site) are probably due to its proximity to populated areas. The main factor provoking degradation in this Mediterranean area was the increase of rural population from the beginning of the 20th century

until late 1950s (Grove and Rackham, 2001). At that time, timber extraction, the use of tussock fiber for textile manufacturing and extensive farming were common economic activities likely increasing anthropic pressure on the "degraded" site. Afterwards, rural exodus during the mid-century involved the abandonment of these agriculture practices. However, although degradation drivers are not currently active, their effects are still observable in the area corresponding to

a case of "relict" degradation (Puigdefábregas and Mendizábal, 2004).

## 2.2 Meteorological and eddy covariance measurements

The net ecosystem-atmosphere exchange of water vapour, $CO_2$ and sensible and latent heat were measured in terms of fluxes via the eddy covariance (EC) technique. Thus, an EC station was installed at each experimental site, AMO and BB (with site codes "Es-Amo" and "Es-Agu" of the European Database Cluster http://www.europe-fluxdata.eu), where ambient and

micrometeorological variables (detailed in Table 2) were monitored continuously since 2009. The EC footprint (i.e. actual measured area) is well within the fetch (i.e. distance to a change in surface characteristics) at both sites. Regarding data processing, the half-hourly averaged fluxes were calculated from raw data collected at 10 Hz using EddyPro 5.1.1 software (Li-Cor, Inc., USA). Flux calculation, flux corrections and quality assessment were performed according to López-Ballesteros et al. (2016).

Additionally, flux measurements acquired under low-turbulence conditions were excluded from the analysis by using a friction velocity ($u_*$) threshold according to the approach proposed by Reichstein et al. (2005). The average $u_*$ thresholds for the whole study period (i.e. 2009–2015) were 0.11 and 0.16 m s$^{-1}$, for AMO and BB, respectively. Furthermore, over the six years of measurements at both sites, data gaps due to low-turbulence conditions, instrument malfunction and theft were unavoidable and not randomly distributed, as noted by Ma et al. (2016). Therefore, the total annual fractions of missing half-

hourly net $CO_2$ fluxes accounted for $33 \pm 3$ % and $29 \pm 6$ % of night-time data and $8 \pm 6$ % and $14 \pm 5$ % of day-time data, for AMO and BB, respectively. Missing data were gap-filled by means of the marginal distribution approach proposed by Reichstein et al. (2005) and uncertainty derived from the gap-filling procedure was calculated by using the variance of the measured data, which was calculated by introducing artificial gaps and repeating the standard gap-filling procedure. Twice the standard deviation of sums of total data was taken as the uncertainty for the several aggregating time periods we used in

the analysis. The annual cumulative C balance was estimated, when possible, by integrating gap-filled half-hourly net $CO_2$ fluxes of good quality (0 and 1 quality flags, according to Mauder and Foken, 2004) over a hydrological year.

In order to test the validity of both EC stations, we assessed the energy balance closure (Moncrieff et al., 1997) by computing the linear regression of half-hourly turbulent energy fluxes, sensible and latent heat fluxes (H+LE; W m$^{-2}$) against available energy, net radiation less the soil heat flux (R$_n$-G; W m$^{-2}$) with the whole six-years database. Storage term in the

soil heat flux was included in the estimates while in case of sensible and latent heat fluxes, this term was negligible given the short height of the vegetation (~50 cm). The resulting slopes were $0.873 \pm 0.002$ (R$^2$ = 0.907) and $0.875 \pm 0.001$ (R$^2$ = 0.920) for AMO and BB, respectively.

**2.3 Flux partitioning**

In order to partition net $CO_2$ ecosystem exchange into Gross Primary Production (GPP) and ecosystem respiration ($R_{eco}$), we firstly modelled the ventilative $CO_2$ efflux by adapting the approach proposed by Pérez-Priego et al. (2013) with the results of previous studies of both sites (López-Ballesteros et al., 2016; 2017). Essentially, we aimed to isolate those moments when

subterranean ventilation ($V_n$) dominates the net $CO_2$ fluxes ($F_c$) and biological fluxes are negligible. These moments correspond to daytime hours during the extremely dry periods. Accordingly, data were selected using the following conditions: (i) net radiation > 10 W m$^{-2}$, (ii) 8 < daily averaged bowen ratio < 10, and (iii) daily soil water content (in bare soil) < 10[th] percentile (in AMO) and < 20[th] percentile (in BB). A less restrictive threshold was used in BB in order to get enough data to build the $V_n$ model, since long-term data gaps occurred at this site during the summer seasons of 2012, 2014

and 2015. Afterwards, in order to build the linear model of $V_n$, these selected $F_c$ data (maximum quality; QC flag=0) were related to the friction velocity ($u_*$).

As the results show (Table 3; Fig. S1), the $V_n$ model is uniquely valid for AMO. Therefore, we only applied the $V_n$ model to AMO data, concretely, during those periods when ventilation (but not exclusively) occurs according to previous research (López-Ballesteros et al., 2017). Hence, the model was applied when: (i) net radiation > 10 W m$^{-2}$, (ii) daily averaged bowen

ratio > 4, (iii) daily soil water content (in bare soil) < 0.01 m$^3$ m$^{-3}$, and (iv) $\sigma_{swc}$ (daily variance of soil water content in bare soil) < 5·10$^{-6}$ (m$^3$ m$^{-3}$)$^2$. We use those moments with very low $\sigma_{swc}$ in order to discern $R_{eco}$ increases caused by rain pulses (Birch effect) from $V_n$ fluxes during the dry season. Then, the modelled ventilative fluxes were subtracted from the measured net $CO_2$ exchange to obtain the $CO_2$ flux corresponding only to biological processes (i.e. biological $F_c$; see Fig. S2).

Finally, the partitioning approach proposed by Lasslop et al. (2010) was applied to the biological $F_c$ for both sites in order to

obtain GPP and $R_{eco}$ fluxes. We chose this approach given the determinant influence of hydric stress, in this case atmospheric drought (assessed via VPD), on the physiology of *Machrocloa tenacissima*, the dominant plant species of the studied semiarid ecosystems (Pugnaire et al., 1996; López-Ballesteros et al. 2016).

**2.4 Enhanced Vegetation Index data series**

We used Enhanced Vegetation Index (EVI) data acquired by the Moderate Resolution Imaging Spectroradiometer (MODIS),

which is on board the Earth Observing System-Terra platform, in order to track vegetation dynamics at both experimental sites. The nominal resolution of EVI products (code "MOD13Q1") is 250 m at nadir and temporal resolution corresponds to 16-day compositing periods. The spatial coordinates used for AMO and BB were N36.8340°, E-2.2526° and N36.9394°, E-2.0341°, respectively.

**2.5 Vadose zone measurements**

Subsoil $CO_2$ molar fraction, temperature and volumetric water content were measured at 0.05 m and 1.50 m below the surface (Table 2) from January 2014 to August 2015 at both experimental sites. In the case of the shallower $CO_2$ sensor, it

was installed vertically with an in-soil adapter (211921GM, Vaisala, Inc., Finland) to avoid water entrance. Subsoil $CO_2$ molar fractions were sampled every 30 s and 5 min averages were stored in a data logger (CR3000 and CR1000, CSI; for AMO and BB, respectively). The deeper $CO_2$ sensor was equipped with a soil adapter for horizontal positioning (215519, Vaisala, Inc., Finland), consisting of a PTFE filter to protect to the $CO_2$ sensor from water. It was buried in the summer of 2013 and the measurements were made every 30 s and stored as 5 min averages in a datalogger (CR1000 and CR23X Campbell Sci., Logan, UT, USA, for AMO and BB, respectively). All $CO_2$ molar fraction records were corrected for variations in soil temperature and atmospheric pressure.

## 2.7 Statistical analysis

All meteorological and soil variables monitored at each site were compared through computation of the non-parametric two-sided Wilcoxon summed rank test in order to detect those factors/variables influencing potentially distinct ecosystem functioning between sites. This test was chosen because variables used satisfied the independence and continuity assumptions but not all were normally distributed. The confidence level used was 95%. The effect size was evaluated using the median of the difference between the samples (AMO minus BB), which was expressed as a standardized value (divided by its standard deviation; $Diff_{st}$; dimensionless) in order to be able to compare results among different variables. This analysis was performed by using three different periods: the whole study period, the period from May to September and the period from May to September during only daytime. These periods were selected given their demonstrated coincidence with high relevance of non-biological processes. All calculations were performed using R software version 3.2.5.

Additionally, in order to include the relationship between pressure and subsoil $CO_2$ variations as a potential factor influencing net $CO_2$ exchange (Sánchez-Cañete et al., 2013), we firstly calculated, separately for each site, Spearman correlation coefficients to determine the time step (6, 12, 24 or 72 hours) with the highest correlation between the differential transformation of pressure and the subsoil $CO_2$ molar fraction at 1.50 m.

## 3 Results

### 3.1 Ambient conditions

Over the study period, the wettest hydrological year was 2009/2010, with annual precipitation of ~500 mm (ca. twice the mean annual precipitation for both sites over the study period, Fig. 2). On the contrary, the driest year was 2013/2014, with annual precipitation of ~100 mm for both sites, less than half the annual average precipitation registered at AMO and BB. Generally, months with precipitation exceeding 20 mm occurred from the beginning of autumn until midwinter, however, in case of 2009/2010, 2010/2011, 2012/2013 and 2014/2015, relevant rain events were registered during spring months. By contrast, in 2013/2014, precipitation was always below 20 mm with the exception of November and December, for both sites, and June, in the case of AMO (Fig. 2a). Commonly, while maximum precipitation usually occurred from November to February, there was a remarkable drought period over summer months (June-August) when it scarcely ever rained (Fig. 2).

Regarding air temperature ($T_{air}$) patterns, monthly averaged $T_{air}$ ranged from 9.6 and 8.1 °C to 27.6 and 27.9 °C in AMO and BB, respectively, over the entire study period. Based on half-hourly averaged data, minimum and maximum $T_{air}$ values registered were 0.1 and 37.9 °C in AMO, and -1.3 and 39.9 °C, in BB, respectively. On one hand, those months with $T_{air}$ above 15 °C usually corresponded to April-November, approximately. Additionally, August was the month with the highest

average $T_{air}$ at both sites, with $T_{air}$ ranges of 25.2 - 27.6 °C at AMO and 24.9 - 27.9 °C at BB, respectively (Fig. 2), over the study period. On the other hand, the lowest monthly average $T_{air}$ usually occurred in January but sometimes also in December and February, with 11.2 – 12.3 °C at AMO and 8.1 - 14.1 °C at BB.

## 3.2 Annual carbon balances

The comparison of the annual C balance among sites was only possible for three hydrological years, 2009/2010, 2010/2011

and 2012/2013, due to long-term data gaps existing in BB during other years. The annual cumulative net $CO_2$ exchange was always positive for AMO (i.e., net C release), whereas BB was neutral or even acted as a C sink over the three years (Fig. 3). For example, in 2009/2010, the net C uptake measured in BB equated to $32 \pm 10$ g C m$^{-2}$ while in AMO, a total amount of $185 \pm 10$g C m$^{-2}$ was released to the atmosphere (Fig. 3a). The year with the largest difference between sites was 2010/2011, with annual C release of $240 \pm 8$ and $-38 \pm 10$ g C m$^{-2}$ in AMO and BB, respectively (Fig. 3b). Likewise, 2012/2013 was the

15 year when the lowest $CO_2$ release was measured in AMO with $163 \pm 7$ g C m$^{-2}$ while a neutral C balance was measured in BB with $0 \pm 8$ g C m$^{-2}$ (Fig. 3c).

Overall, a positive and saturating trend was observed at both sites during autumn months until December-February when cumulative net $CO_2$ releases start to decline. The autumn net $CO_2$ release (i.e., positive values) was usually higher in AMO than in BB, excepting for 2012/2013, and the declining slope was always higher in BB, meaning greater net C uptake rates.

Although the pattern of the cumulative net $CO_2$ exchange showed differences between sites over autumn, winter and spring months, stronger discrepancies were found during summer droughts. Concretely, from April-May until August, BB showed neutral behavior while a remarkable positive trend was observed in AMO, denoting a large net $CO_2$ release.

## 3.3 Seasonal and diurnal net $CO_2$ exchanges

Long-term data loss occurred in BB during the springs of 2011/2012, 2013/2014 and 2014/2015 and summers of 2013/2014

and 2014/2015, when annual C balances could not be estimated. However, by observing the available seasonal data, it is noticeable that, maximum and minimum seasonal net $CO_2$ exchanges were very different between sites (Fig. 4). On one hand, maximum seasonal net $CO_2$ uptake was measured during winter (December-February) in AMO and over spring (March-May) in BB, when peaking net $CO_2$ uptake fluxes equated to -31 g C m$^{-2}$ (winter 2011/2012) and -105 g C m$^{-2}$ (spring 2010/2011) in AMO and BB, respectively. Additionally, net $CO_2$ uptake was only observed during three winters in

the case of AMO, whereas it was frequently measured during both winter and spring in BB. On the other hand, cumulative net $CO_2$ release to the atmosphere occurred over all seasons in AMO, but acutely in summer, when maximum seasonal net

$CO_2$ release was always observed ranging from 111 to 153 g C m$^{-2}$. In contrast, in BB, the highest $CO_2$ effluxes usually occurred in autumn ranging from 25 to 74 g C m$^{-2}$, although significant $CO_2$ release was also observed in winter 2013/2014 and the summers of 2009/2010-2011/2012. Regarding seasonal evapotranspiration (ET) fluxes, results showed a ~30% higher ET at BB compared to AMO during spring. Major inter-site differences in autumn occurred in the first and last year of study, when ET was 23% and 12% higher at BB, respectively (Fig. 5).

Comparing daily-scale net $CO_2$ exchange and ET fluxes with Enhanced Vegetation Index (EVI) data, we can notice some similarities in the general patterns of both sites (Fig. 6). Roughly, there was a common annual pattern in which the highest values of EVI coincided with maximum net $CO_2$ uptake rates (i.e. negative net $CO_2$ fluxes), which in turn, corresponded to peaking ET fluxes. Additionally, a decreasing trend in EVI over the 6 years of study was also noticeable for both sites. However, some inter-site and inter-annual differences were evident (Fig. 6).

On one hand, there were two main differences between sites. Firstly, extreme net $CO_2$ release was measured uniquely in AMO during summer months (June-August), when maximum net $CO_2$ fluxes ranging from 31 to 68 g C m$^{-2}$ were measured (Fig. 6b). Over the study period, the monthly net $CO_2$ exchange of AMO during dry seasons was up to one hundred times higher than in BB (in August 2013), since monthly net $CO_2$ fluxes measured in BB were much lower, from -8 to 16 g C m$^{-2}$ (Fig. 6b). Besides the striking differences in summer net $CO_2$ exchange between sites, minor discrepancies were also found in ET fluxes and EVI for the same drought periods. In this regard, monthly averaged ET over the dry season equated to 13 ± 4 and 10 ± 4 mm for AMO and BB, respectively, and EVI was on average 4% higher in BB than in AMO (Fig. 6a, c). The second inter-site difference was the greater net $CO_2$ uptake over longer periods measured in BB. Concretely, the period during which the ecosystems acted as C sinks lasted on average 38 days longer in BB than in AMO annually (Table 4). Accordingly, the annual amount of C fixation ranged from 6-59 g C m$^{-2}$ at AMO and 15-129 g C m$^{-2}$ at BB, respectively, with the annual averaged net C uptake in BB 162% higher than at AMO (Table 4). Consequently, peaking EVI values were usually observed during March-April for both sites, however, over winter and spring months (growing period), EVI measured at BB was 3 - 37% higher than AMO, with the largest inter-site differences in 2009/2010 and 2014/2015 (Fig. 6a). Likewise, monthly averaged ET fluxes measured at BB over winter and spring months (December-May) were from 3 to 24% larger than those measured at AMO. Additionally, the growing period of the driest year (2013/2014) corresponded to the lowest monthly ET fluxes and the least difference between sites.

On the other hand, differences in the inter-annual variability of EVI were found between years. Concretely, 2009/2010 and 2013/2014 were the years with maximum and minimum annual precipitation and EVI observations, respectively, for both sites. In 2009/2010, EVI observations were 28% and 20% higher than the six-year averaged values in BB and AMO, respectively. In case of the driest year, 2013/2014, growing season (winter-spring) EVI was reduced 35% and 28% in BB and AMO, respectively. Nevertheless, the largest difference between sites in winter-spring EVI observations was found in 2014/2015, following the driest year, when BB showed a pattern very similar to those registered over the years prior to the dry spell, while AMO still presented EVI values 21% below the six-year average (Fig. 6a).

### 3.4 Biological net $CO_2$ exchange, gross primary production, ecosystem respiration and water use efficiency

The results of the "biological" annual C balance, which was obtained, in case of AMO, by applying the ventilation model, are in accordance with our hypotheses. Annual C emission was always measured at AMO, whereas BB acted as a neutral and mild C sink. On average, AMO emitted 32 g C m$^{-2}$ more than BB. At a monthly time scale, net $CO_2$ fluxes during autumn were, on average, ~4 times higher at BB, excepting the last study year, when the net $CO_2$ emission at AMO was 21 times greater than at the "natural" site. However, during winter and spring months, net $CO_2$ uptake was generally higher at the BB (Fig. 7a).

On average, during the six years of study, GPP, $R_{eco}$ and Water Use Efficiency (WUE) were nine, two and ten times higher, respectively, at BB compared to AMO. Firstly, GPP was always higher at BB compared to AMO (Fig. 7c). Major differences occurred in autumn 2014/2015, when monthly cumulative GPP at BB was 32 times higher on average. Similarly, $R_{eco}$ was generally higher, up to ~8 times (October 2014), at BB. However, respiratory fluxes were occasionally greater at AMO, from 2% to 31% higher, during spring and winter months of all studied years excepting 2013/2014 (Fig. 7b). Maximum inter-site differences in GPP and $R_{eco}$ were found in winter and autumn 2014/2015, following the driest year, when monthly GPP was ~30 times higher at BB compared to AMO. Similarly, monthly $R_{eco}$ was ~5 times greater at BB. Inter-site differences in partitioned fluxes could not be assessed during spring months due to the lack of data from BB. Secondly, WUE was lower at AMO during the whole study period, when maximum and minimum differences coincided with the highest and lowest differences in GPP between sites. On average, monthly WUE was 6 and 1.5 times higher at BB during winter and spring seasons. Major inter-site differences were found in autumn and winter 2014/2015 (Fig. 7d).

### 3.5 Differences in meteorological and soil variables between sites

Results from the two-sided Wilcoxon summed rank test (Table 5) showed significant differences (p-value < 0.05) between sites in most of the monitored meteorological variables. The few exceptions were the friction velocity (u$_*$), when using the whole study period, the maximum wind speed registered every half-hour (WS$_{max}$), when analyzing May-September data, and the wind speed (WS) and precipitation when assessing daytime May-September data (Table 5). The great amount of observations (n ranged from 21410 to 205751) produced highly significant results (Table 5). Hence, the standardized difference between the samples (Diff$_{st}$) allowed us to quantitatively explore the differences between sites. Relevant differences (Diff$_{st}$>1) were found only for the subsoil $CO_2$ molar fraction measured at 1.50 m depth ($CO_{2,\,1.50m}$) for all periods, and during May-September months even when using only daytime data (Table 6). Concretely, $CO_{2,\,1.50m}$ was always higher in AMO, from 889 to 1109 ppm (Table S1, S2 and S3). Additionally, volumetric water content at 0.05m depth (VWC$_{0.05m}$) was also higher in AMO compared to BB but only during summer months (Table 6), when absolute differences were very small, ranging from 0.028 to 0.037 m$^3$ m$^{-3}$ (Tables S1, S2 and S3). In contrast, subsoil $CO_2$ molar fraction measured at 0.05 m depth ($CO_{2,\,0.05m}$) was from 89 to 150 ppm higher in BB when analyzing dry season (May-September) daytime data (Table 6, S1, S2 and S3).

The temporal dynamics of subsoil $CO_2$ molar fractions revealed similar annual patterns between sites; generally however, $CO_{2, 0.05m}$ was higher in BB, from 6 to 88%, while $CO_{2, 1.50m}$ was always greater in AMO, from 31 to 97% (Fig. 8). On one hand, the maximum monthly averaged values of $CO_{2, 0.05m}$ were registered in autumn, concretely, in November and October with 642 and 1120 ppm in AMO and BB, respectively, whereas minimum values occurred in September and August with 373 and 400 ppm at each site (Fig. 8a). On the other hand, peak monthly averaged values of $CO_{2, 1.50m}$ occurred in July for both sites, with 2751 and 1602 ppm in AMO and BB, respectively, although relatively high $CO_{2, 1.50m}$ was also measured during November in BB. On the contrary, minimum values were observed in December and February, with 1364 and 735 ppm in AMO and BB, respectively (Fig. 8b).

Finally, results of the Spearman correlation analysis between net $CO_2$ exchange and belowground $CO_2$ at 1.50 m depth ($CO_{2, 1.50m}$) showed a positive relationship between both absolute variables, which was stronger in case of AMO compared to BB with Spearman correlation coefficients ($r_s$) of 0.30 and 0.11, respectively (Table 7). In contrast, a negative relationship was found between pressure and $CO_{2, 1.50m}$ with higher $r_s$ in AMO compared to BB (Table 7). Additionally, $r_s$ were maxima at 12 h intervals for $CO_{2, 1.50m}$ and pressure increments ($dP_{12h}$) at AMO, and at 6 h intervals for $CO_{2, 1.50m}$ and pressure increments ($dP_{6h}$, respectively) at BB, with $r_s$ equal to -0.87 and -0.63, respectively.

## 4 Discussion

Our results verify that land-degradation affects the C sequestration capacity of semiarid ecosystems, since relevant differences between sites were observed during the growing season, when greater net C uptake over longer periods was observed in the "natural" site (BB). However, contrary to what we previously hypothesized, much greater net C release was measured at the "degraded" site (AMO) over drought periods due to the predominance of subterranean ventilation (López-Ballesteros et al., 2017). In fact, the great difference in annual C budgets between sites (Fig. 3) was largely related to this process resulting in an average release of 196 ± 40 and -23 ± 20 g C m$^{-2}$ yr$^{-1}$ for the "degraded" and "natural" sites, respectively. Even when assessing only the biological net $CO_2$ exchange, by subtracting the non-biological $CO_2$ flux when feasible, the "degraded" site emitted 32 g C m$^{-2}$ more than the "natural" site. In this regard, the ecosystems' functioning could be divided into three different phases. The first phase corresponded to the autumn months, when the first rainfall events after the dry summer (i.e. rain pulses) activated the soil microbiota triggering respiratory $CO_2$ emissions as previously measured at the same experimental sites (López-Ballesteros et al., 2016; Rey et al., 2017). During this phase, maximum net $CO_2$ emission was observed at the "natural" site, which exceeded the biological net $CO_2$ flux observed in the "degraded" site (Fig. 7a). In fact, as hypothesized, $R_{eco}$ was generally higher at the "natural" site during autumn months (Fig. 7b) probably due to the greater pool of organic carbon (Table 1) and the potential differences in microbial communities (Rey et al., 2017) between sites. The second phase comprised the growing period, when plants photosynthesized and also respired along with microorganisms under milder temperatures and better hydric conditions. During this phase, larger net $CO_2$ uptake was measured at the "natural" site, concretely 162% more than in the "degraded" site (Table 4) due to the higher vegetation cover

and more fertile soils (Table 1) of the "natural" site. Accordingly, GPP estimates were, on average, nine times higher at the "natural" site (Fig. 7c) and moreover, these results were supported by the lower ET, WUE and EVI values obtained in the "degraded" site during winter and spring months over the whole study period (Fig. 5, 6 and 7d). The third phase consisted of the dormancy period when water scarcity and high temperatures constrained biological activity. During this period, a neutral

C balance was observed at the "natural" site while extreme $CO_2$ release was measured at the "degraded" site, as expected (Fig. 4, 6b), where ventilative fluxes were dominant.

In order to detect potential factors driving the observed differences in the C balances, we checked whether soil and meteorological variables differed between sites. Our results demonstrated that some factors typically influencing GPP$R_{eco}$, and hence net ecosystem $CO_2$ exchange, such as photosynthetic photon flux density (PPFD; Michaelis and Menten, 1913),

precipitation (Berner et al., 2017; Jongen et al., 2011), vapor pressure deficit (VPD; Lasslop et al., 2010) and soil and air temperature (Lloyd and Taylor, 1994), did not differ between sites (Table 5, 6). Conversely, some differences were found for shallow soil volumetric water content ($VWC_{0.05m}$) during dry seasons (Table 6), when $VWC_{0.05m}$ was two times higher in AMO than in BB, but absolute differences were slight, from 0.028 to 0.037 $m^3$ $m^{-3}$ (Tables S1,S2 and S3). Hence, although the important influence of soil moisture in both GPP and $R_{eco}$ is known (Tang and Baldocchi, 2005), we believe that

differences in $VWC_{0.05m}$ are not enough relevant to cause the differing ecosystem functioning observed over the drought period. Additionally, we think that this inter-site difference in $VWC_{0.05m}$ could be instrumental, or due to the spatial variability of $VWC_{0.05m}$ derived from the heterogeneity of soil morphological characteristics, since we only used one sensor at each site. Similarly, important differences were not detected in several variables linked to subterranean ventilation, such as the friction velocity (u$_*$; Kowalski et al., 2008), wind speed (WS; Rey et al., 2012), half-hourly maximum wind speed

($WS_{max}$) and net radiation, which has been positively correlated to ventilative $CO_2$ fluxes (López-Ballesteros et al., 2017), when using the analysis periods when this process is supposed to be relevant, namely daytime hours during the dry seasons (Table 5). However, although no turbulence and wind speed inter-site differences were found, interconnectivity of soil pores and fractures is probably higher at the "degraded" site (Table 1) due to its higher gravel and rock fractions (Table 1), which could lead to an enhanced penetration of eddies within the vadose zone (Pérez-Priego et al., 2013).

Apart from that, outstanding differences between sites were observed in subsoil $CO_2$ molar fractions measured at 0.05 and 1.50 m depths ($CO_{2,\ 0.05m}$ and $CO_{2,\ 1.50m}$, respectively; Table 6). On one hand, $CO_{2,\ 0.05m}$ was generally higher in the "natural" site given its lower degradation level, which probably promotes a higher microbial activity supported by higher vegetation density and soil fertility (Table 1) especially during spring (Fig. 8), as pointed also by Oyonarte et al. (2012). On the other hand, $CO_{2,\ 1.50m}$ values were acutely higher in the "degraded" site, by up to 1000 ppm compared to the "natural" site (Tables

S1, S2 and S3). Therefore, we suggest that $CO_{2,\ 1.50m}$ is the main factor responsible for the inter-site differences in net $CO_2$ fluxes over the dry season. In this regard, previous research has suggested two potential origins of this vadose zone $CO_2$, geological degassing (Rey et al. 2012b) and/or subterranean translocation of $CO_2$ in both gaseous and aqueous phases

(López-Ballesteros et al. 2017). However, not only the amount of subsoil $CO_2$ matters but also how effective is its transport, since both determine the net $CO_2$ release from the vadose zone to the atmosphere. In this context, Oyonarte et al. (2012) found, in the same study area (Cabo de Gata-Níjar Natural Park), that soils with degradation symptoms, such as lower SOC, depleted biological activity, coarser texture and worse structure, showed higher soil $CO_2$ effluxes over the dry season.

Additionally, soil $CO_2$ effluxes measured during summer months correlated positively with the fraction of rock outcrops, suggesting that deteriorated soil physical conditions actually enhanced vertical transfer of $CO_2$-rich air from subsoil to the atmosphere (Oyonarte et al., 2012). In fact, correlation analysis between $CO_{2,\ 1.50m}$ and net $CO_2$ exchange/atmospheric pressure (Table 7) showed a stronger relationship between these variables at the "degraded" site. In this sense, ecosystem degradation could provoke a greater exposure of subsoil $CO_2$ to the pressure effect, as described by Sánchez-Cañete et al.

(2013), probably due to a higher fraction of bare soil, coarser structure, differing porosity type and/or thinner soil depth (Table 1).

Regarding EVI data, we found a discrepancy between GPP estimates and EVI values since, contrary to what is observed in EVI results, we observed that GPP was always higher at the "natural" site compared to the "degraded" site. We think that this is due to the different spatial scales defining each measurement. MODIS pixels have an area of ~6.25 ha while the eddy

covariance footprint corresponds to a smaller area of ~1 ha. Therefore, there is an EVI uncertainty that stems from the influence of other surface elements apart from vegetation, such as bare soil or outcrops within the pixel, which is our case. In fact, previous studies confirm the discrepancy between MODIS- and EC-derived GPP estimates, especially on sparse vegetation areas with low productivity (Gilabert et al., 2015). However, EVI data have allowed us to complement our findings based on $CO_2$ fluxes, especially when EC data losses occurred. For instance, the declining trend observed from

2009/2010 until the end of the study period, for both sites, was not noticeable from EC data alone (Fig. 6). This long-term decrease in EVI may be related to a gradual drying following the wettest year (2009/2010), when extraordinarily high precipitation (twice the mean annual precipitation for both sites over the study period) occurred. This EVI pattern also suggests a pulse-like behaviour of ecosystem vegetation over the inter-annual time scale.

Moreover, in addition to demonstrating that degradation can influence the biological activity of ecosystems' vegetation, EVI

results also showed that degradation level can modulate how an ecosystem responds to a short-term disturbance. A clear example is the dry spell experienced in 2013/2014, when a reduction in EVI was measured during the growing season in both sites, i.e. 35% and 28% in the "natural" and "degraded" sites, respectively. However, a year later (2014/2015), EVI values below the six-year average were observed only at the "degraded" site (21% lower; Fig. 6a) and major inter-site differences were found for GPP, $R_{eco}$ and WUE during autumn and winter months (Fig. 7b, 7c and 7d). Accordingly, the

"natural" site seemed to be more resilient than the "degraded" site against the short-term disturbance, since the effect of drought persisted in AMO even during the following year, while BB recovered to a pre-perturbation state within the same period (Fig. 6a). As a result, ecosystem resilience (Holling, 1973) was lessened by long-term disturbances such as land degradation, making degraded ecosystems more vulnerable to climate extremes (Reichstein et al., 2013). In this sense,

mitigation policies to confront land degradation should be focused on prevention programs since ecosystem restoration does not recover complete ecosystem functionality (Lal, 2001;Moreno-Mateos et al., 2017). Moreover, even after several decades, relict degradation legacies can remain (Alados et al., 2011).

## 5 Conclusions

The present study can be seen as a step forward to better understanding the effect of land degradation on the intricate network of multi-scale processes, factors and structures that define ecosystems' biological productivity and ultimately control their C balances. Despite some limitations, such as long-term data gaps, this research demonstrates that continuous ecosystem-scale EC observations remain crucial to comprehend how climate and land use change can modify the C sequestration capacity of ecosystems. In fact, annual average release of $196 \pm 40$ and $-23 \pm 20$ g C m$^{-2}$ yr$^{-1}$ for the "degraded" and "natural" (i.e. site of reference) sites were measured, respectively. Additionally, larger net $CO_2$ uptake over longer periods was observed at the "natural" site, concretely an amount of C 162% higher compared to the "degraded" site, whereas much greater net $CO_2$ release was measured at the "degraded" site during drought periods. Accordingly, the estimates of gross primary production, ecosystem respiration and water use efficiency were, on average, nine, two and ten times higher in the "natural" site, respectively. Future research should be based on the continuity of long-term monitoring stations, such as eddy covariance stations, in order to calibrate and validate satellite data, reduce uncertainties in the relationships between ecosystem productivity, land degradation and climate change and finally, to improve the predictive ability of current terrestrial C models.

## Data availability

The eddy covariance data are available in the European Database Cluster (http://www.europe-fluxdata.eu) where experimental sites have the codes "Es-Amo" and "Es-Agu". Other data can be obtained by contacting the corresponding author.

## Author contribution

FD, CO, AK and PSO designed the experiment. ALB, PSO, EPSC and MRM calibrated the sensors, collected the data and maintained the field instrumentation. EPSC designed subsoil data acquisition system and MRM processed subsoil data. ALB processed the eddy covariance data, made the figures and tables and wrote the manuscript. All authors reviewed the manuscript.

**Acknowledgements**

A. López-Ballesteros acknowledges support from the Spanish Ministry of Economy and Competitiveness (FPI grant, BES-2012-054835). This work was supported in part by the Spanish Ministry of Economy and Competitiveness projects ICOS-SPAIN (AIC10-A-000474), SOILPROF (CGL2011-15276-E), GEI-Spain (CGL2014-52838-C2-1-R), including European Union ERDF funds; and by the European Commission project DIESEL (PEOPLE-2013-IOF-625988). We also thank L. Morillas, O. Uclés and E. Arnau for field work.

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

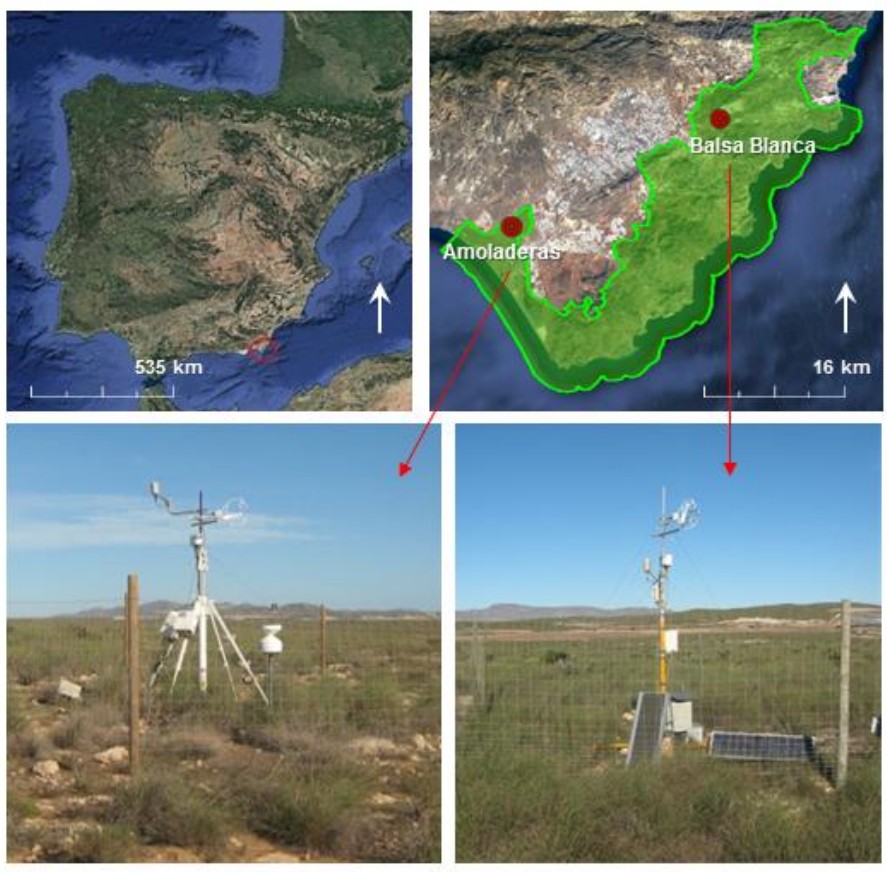

**Figure 1: Location (above) and photographs (below) of the experimental sites. Green area represents the Cabo de Gata-Níjar Natural Park (Almeria, Spain).**

**Table 1: Site characteristics, surface fractions and soil properties of both experimental sites studied. Asterisks denote significant differences (p-value<0.05). Adapted from Rey et al., (2011).**

| | Amoladeras | Balsa Blanca |
|---|---|---|
| *Surface fractions* | | |
| Vegetation cover (%) | 23.1 ± 2.4* | 63.2 ± 5.2* |
| Litter (%) | 10.5 ± 2.0 | 8.1 ± 1.9 |
| Biological crust (%) | 23.1 ± 2.8 | 18.2 ± 3.8 |
| Bare soil (%) | 8.1 ± 0.1* | 0.3 ± 0.3* |
| Gravel (%) | 21.1 ± 0.1* | 8.6 ± 2.5* |
| Rock (%) | 14.0 ± 1.2* | 1.5 ± 0.5* |
| *Soil properties* | | |
| Soil type | Lithic Leptosol (Calcaric) | Mollic Lithic Leptosol (Calcaric) |
| Maximum soil depth (cm) | 10 | 20 |
| Soil texture class | Sandy loam | Sandy loam |
| Clay (%) | 14.6 | 16.1 |
| Silt (%) | 27.0 | 22.8 |
| Sand (%) | 58.4 | 61.1 |
| Bulk density (g cm$^{-3}$) | 1.11± 0.04 | 1.25 ± 0.09 |
| SOC (kg m$^{-2}$) | 1.24 | 4.64 |
| Carbonates (%) | 14 | 2 |

**Table 2: Variables measured, sensors used and their installation height in Amoladeras and Balsa Blanca experimental sites.**

| Variable | Sensor | Sensors height | |
|---|---|---|---|
| | | Amoladeras | Balsa Blanca |
| *Eddy Covariance system* | | | |
| Wind speed (3-D) and sonic temperature | A three-axis sonic anemometer (CSAT-3, Campbell Scientific Inc, Logan, UT, USA; hereafter CSI) | 3.05 m | 2.90 m |
| $CO_2$ and $H_2O$ vapour densities | A open-path infrared gas analyzer (Li- 7500, Li-Cor, Lincoln, NE, USA) | 3.05 m | 2.90 m |
| *Meteorological and soil measurements* | | | |
| Air pressure | A open-path infrared gas analyzer (Li-Cor 7500, Lincoln, NE, USA) | 1.60 m | 1.80 m |
| Photosynthetic Photon Flux Density | Two PAR sensors (Li-190, Li-Cor, Lincoln, NE, USA) | 1.40 m | 1.50 m |
| Net Radiation | A net radiometer (NR Lite, Kipp&Zonen, Delft, Netherlands) | 1.70 m | 1.50 m |
| Air temperature | A thermohygrometer (HMP35-C, CSI) | 3.62 m | 1.50 m |
| Air relative humidity | A thermohygrometer (HMP35-C, CSI) | 3.62 m | 1.50 m |
| Subsoil Water Content | Two water content reflectometers (CS616, CSI) | -0.05 and -1.50 m | -0.05 and -1.50 m |
| Subsoil temperature | Two soil temperature probes (TCAV, CSI) | -0.05 and -1.50 m | -0.05 and -1.50 m |
| Subsoil $CO_2$ molar fraction | A $CO_2$ sensor (GMP-343, Vaisala, Inc., Finland) | -0.05 m | -0.05 m |
| Subsoil $CO_2$ molar fraction | A $CO_2$ sensor (GMM222, Vaisala, Inc., Finland) | -1.50 m | -1.50 m |
| Rainfall | A tipping bucket (0.2 mm) rain gauge (785 M, Davis Instruments Corp., Hayward, CA, USA) | 1.30 m | 1.40 m |

5    **Table 3: Linear regression results between half-hourly net $CO_2$ fluxes of maximum quality (QC flag=0) and friction velocity ($u_*$) used to model subterranean ventilation.**

| Model parameters | Amoladeras | Balsa Blanca |
|---|---|---|
| Intercept ± error (p-value) | -1.876 ± 0.291 (4e-09) | 0.628 ± 0.508 (0.226) |
| Slope ± error (p-value) | 8.500 ± 0.549 (<2e-16) | 0.578 ± 0.944 (0.545) |
| $R^2$ | 0.706 (<2.2e-16) | 0.013 (0.5451) |
| n | 102 | 31 |

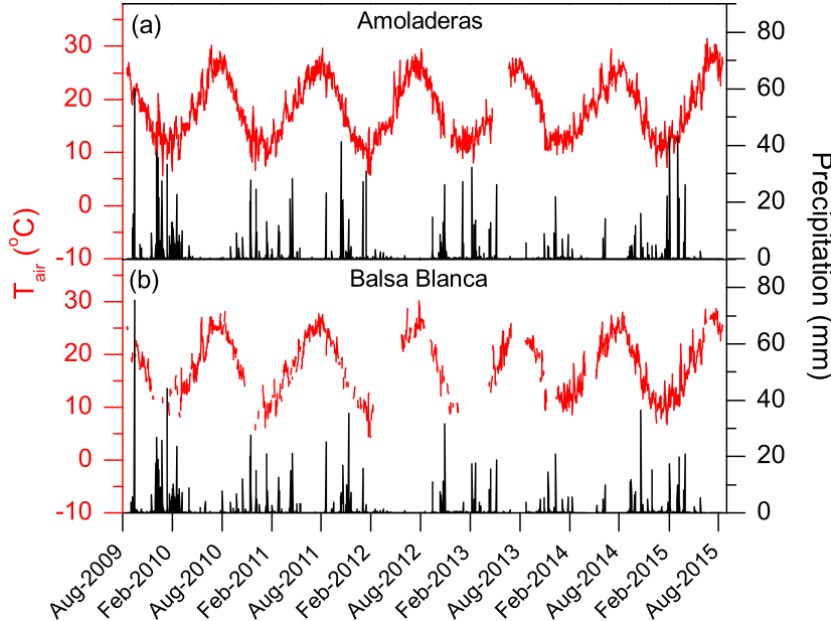

**Figure 2: Daily averages of air temperature ($T_{air}$) and precipitation in (a) Amoladeras and (b) Balsa Blanca.**

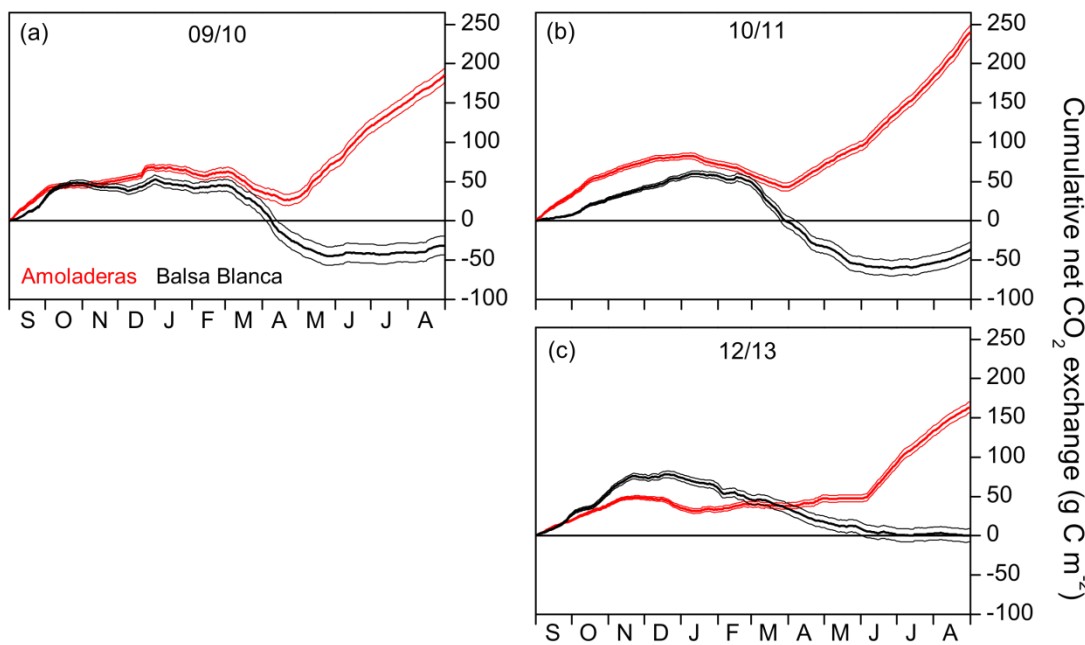

**Figure 3: Cumulative annual net $CO_2$ exchange over the three hydrological years without long-periods of missing data in both experimental sites, Amoladeras (red lines) and Balsa Blanca (black lines). Negative values denote net carbon uptake while positive values denote net carbon release. Thin lines indicate uncertainty derived from the gap-filling procedure.**

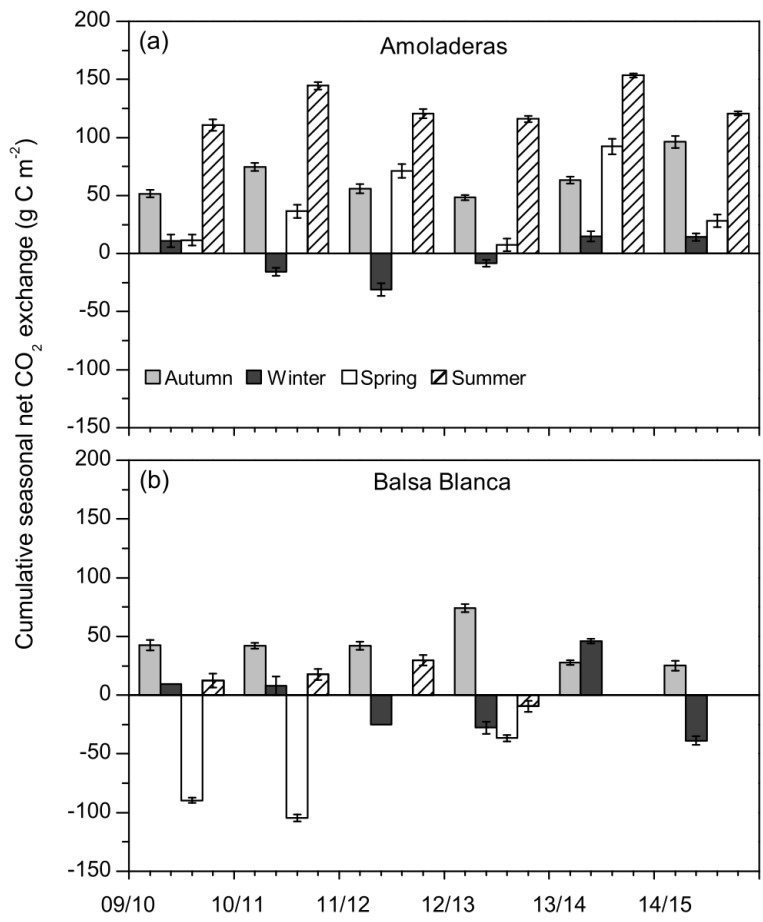

**Figure 4: Cumulative seasonal net CO$_2$ exchange over the study period in both experimental sites. Negative values denote net carbon uptake while positive values denote net carbon release. In case of Balsa Blanca, lacking bars correspond to long-term data losses (>50% data).**

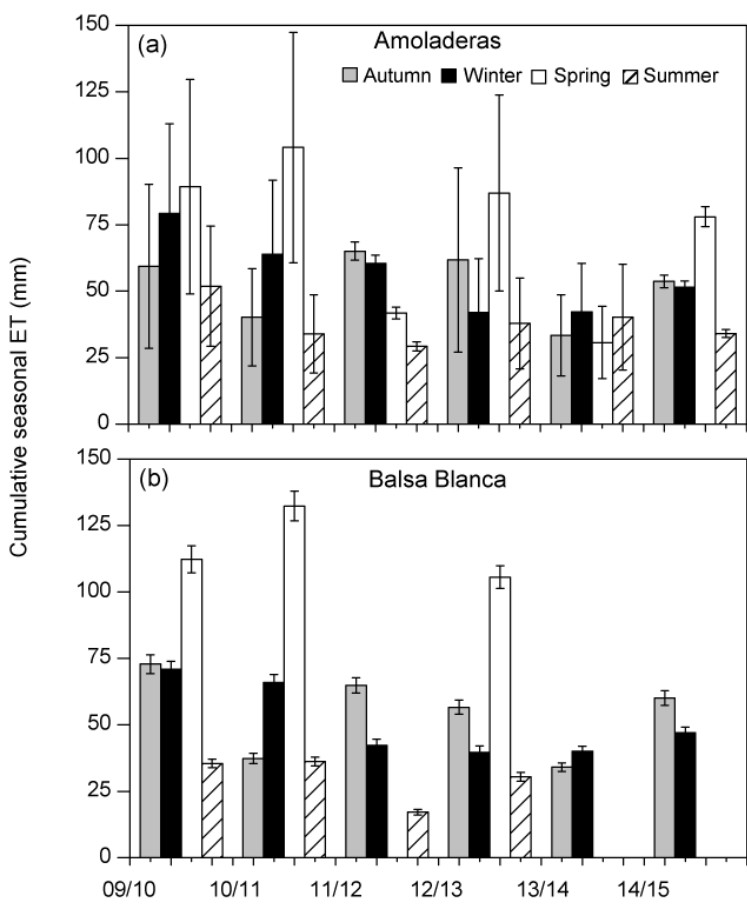

**Figure 5: Cumulative seasonal evapotranspiration fluxes (ET) over the study period in both experimental sites. In case of Balsa Blanca, lacking bars correspond to long-term data losses (>50% data). Error bars denote uncertainty derived from the gap-filling procedure.**

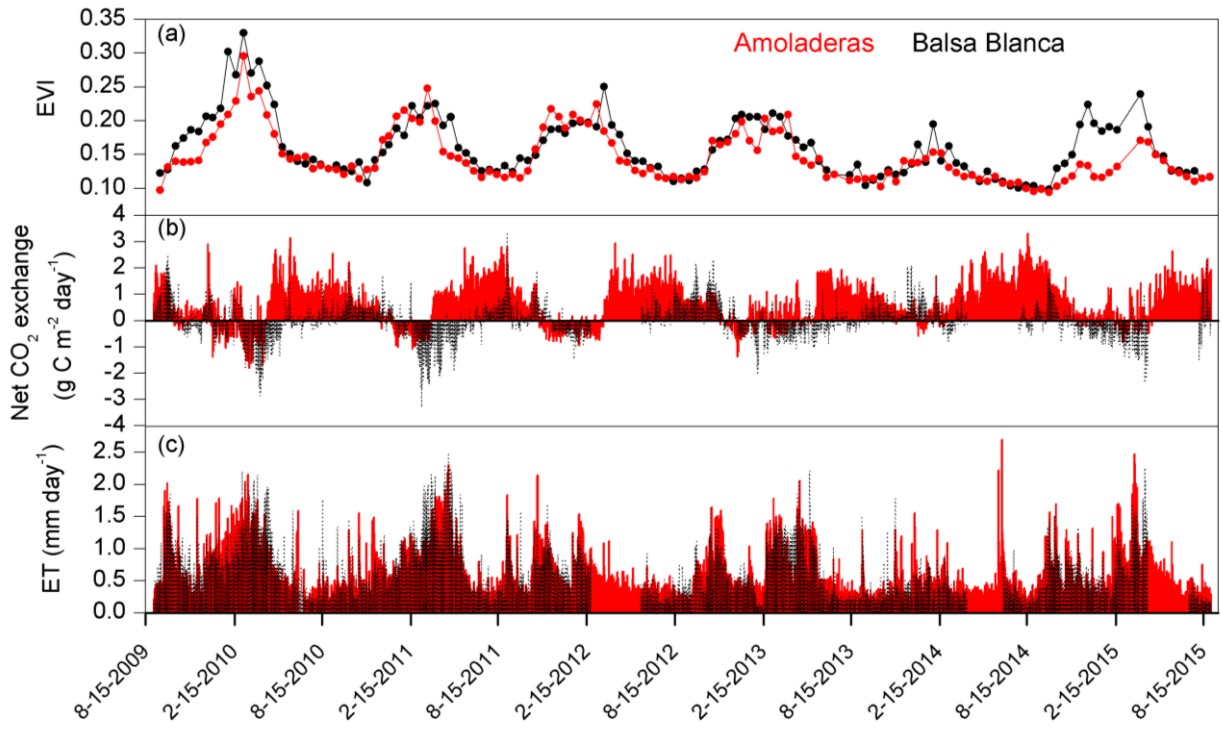

**Figure 6: Time series of (a) Enhanced Vegetation Index (EVI), (b) daily net CO₂ exchange and (c) daily evapotranspiration fluxes measured in Amoladeras (red lines and dots) and Balsa Blanca (black lines and dots) over six hydrological years (2009-2015). Long-term data losses correspond to periods of several months when ET and CO₂ fluxes are absent.**

**Table 4: Number of days with daily net CO₂ uptake and the related total C absorbed for every hydrological year and every field site of the study. Asterisks denote those years with abundant data losses (~30% data).**

| Site | Amoladeras | | Balsa Blanca | |
|------|------------|---|--------------|---|
| Year | N. days of net $CO_2$ uptake | Total net $CO_2$ uptake (g C m$^{-2}$) | N. days of net $CO_2$ uptake | Total net $CO_2$ uptake (g C m$^{-2}$) |
| 09/10 | 58 | $-59 \pm 7$ | 196 | $-125 \pm 12$ |
| 10/11 | 86 | $-41 \pm 4$ | 160 | $-129 \pm 10$ |
| 11/12 | 114 | $-43 \pm 5$ | 104* | $-40 \pm 6$* |
| 12/13 | 103 | $-31 \pm 4$ | 212 | $-96 \pm 8$ |
| 13/14 | 31 | $-6 \pm 12$ | 64* | $-15 \pm 5$* |
| 14/15 | 59 | $- 14 \pm 3$ | 172* | $-103 \pm 9$* |

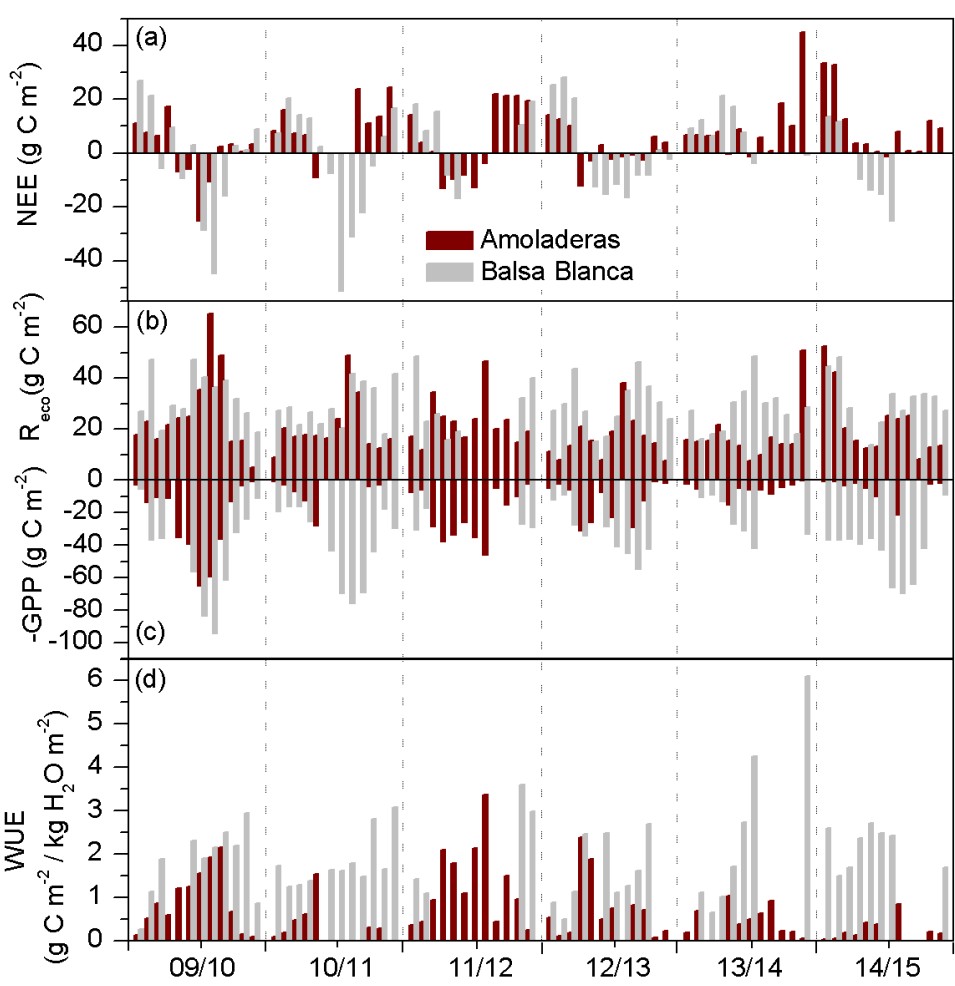

**Figure 7: Monthly cumulative fluxes of (a) biological net ecosystem CO$_2$ exchange, (b) ecosystem respiration (R$_{eco}$), (c) negative gross primary production and (d) water use efficiency over the six hydrological years of study (2009-2015) for Amoladeras (dark red) and Balsa Blanca (grey). Lacking bars correspond to long-term data losses.**

**Table 5: Results of the two-sided Wilcoxon summed rank test used to assess differences among meteorological variables measured at each experimental site over all periods, from May to September and from May to September during daytime, separately. Medians of the difference between the samples (Amoladeras minus Balsa Blanca) in standarized terms (Diff$_{st}$) and number of observations are detailed. Significant results (p-value<0.05) are denoted with asterisks, and bold values represent those variables with Diff$_{st}$ between sites above 1.**

| Variables | All periods | | May - September | | May - September Daytime | |
|---|---|---|---|---|---|---|
| | Diff$_{st}$ | n | Diff$_{st}$ | n | Diff$_{st}$ | n |
| PPFD ($\mu$mol m$^{-2}$ s$^{-1}$) | 0.0009* | 205751 | 0.0009* | 84491 | 0.1378* | 38963 |
| Net radiation (W m$^{-2}$) | -0.0457* | 197924 | -0.0476* | 81019 | -0.1205* | 38963 |
| T$_{air}$ (°C) | 0.0310* | 182240 | 0.1935* | 77866 | 0.0502* | 37480 |
| VPD (hPa) | 0.0783* | 166918 | 0.1370* | 71474 | -0.0938* | 34430 |
| RH (%) | -0.1636* | 197649 | -0.1031* | 80950 | 0.1784* | 38935 |
| u$_*$ (m s$^{-1}$) | -0.0054 | 166346 | -0.0563* | 71194 | -0.1340* | 34284 |
| WS (m s$^{-1}$) | 0.1628* | 166359 | 0.0793* | 71195 | 0.0165 | 34285 |
| WS$_{max}$ (m s$^{-1}$) | 0.1001* | 165458 | 0.0124 | 70635 | -0.0796* | 33994 |
| Pressure (hPa) | 0.3737* | 166336 | 0.5828* | 71188 | 0.5602* | 34280 |
| Precipitation (mm) | -1.95E-05* | 204892 | -4.84E-05* | 83860 | 5.32E-05 | 38963 |

**Table 6: Results of the two-sided Wilcoxon summed rank test used to assess differences among soil variables measured at each experimental site over over all periods, from May to September and from May to September during daytime, separately. Medians of the difference between the samples (Amoladeras minus Balsa Blanca) in standarized terms (Diff$_{st}$) and number of observations are detailed. Significant results (p-value<0.05) are denoted with asterisks, and bold values represent those variables with Diff$_{st}$ between sites above 1.**

| Variables | All periods | | May - September | | May - September Daytime | |
|---|---|---|---|---|---|---|
| | Diff$_{st}$ | n | Diff$_{st}$ | n | Diff$_{st}$ | n |
| CO$_{2, 0.05m}$ (ppm) | -0.4027* | 46340 | -0.6578* | 21413 | **-1.1396***  | 9816 |
| CO$_{2, 1.50m}$ (ppm) | **1.1196*** | 50133 | **1.3517*** | 24347 | **1.3062*** | 11385 |
| T$_{0.05m}$ (°C) | 0.0927* | 46337 | -0.1160* | 21410 | -0.2119* | 9813 |
| T$_{1.50m}$ (°C) | 0.1476* | 50137 | -0.0591* | 24350 | -0.0834* | 11385 |
| VWC$_{0.05m}$ (m$^3$ m$^{-3}$) | 0.8265* | 52353 | **1.2724*** | 25231 | **1.2839*** | 11303 |
| VWC$_{1.50m}$ (m$^3$ m$^{-3}$) | -0.8385* | 53865 | 0.0674* | 24570 | 0.0547* | 11462 |

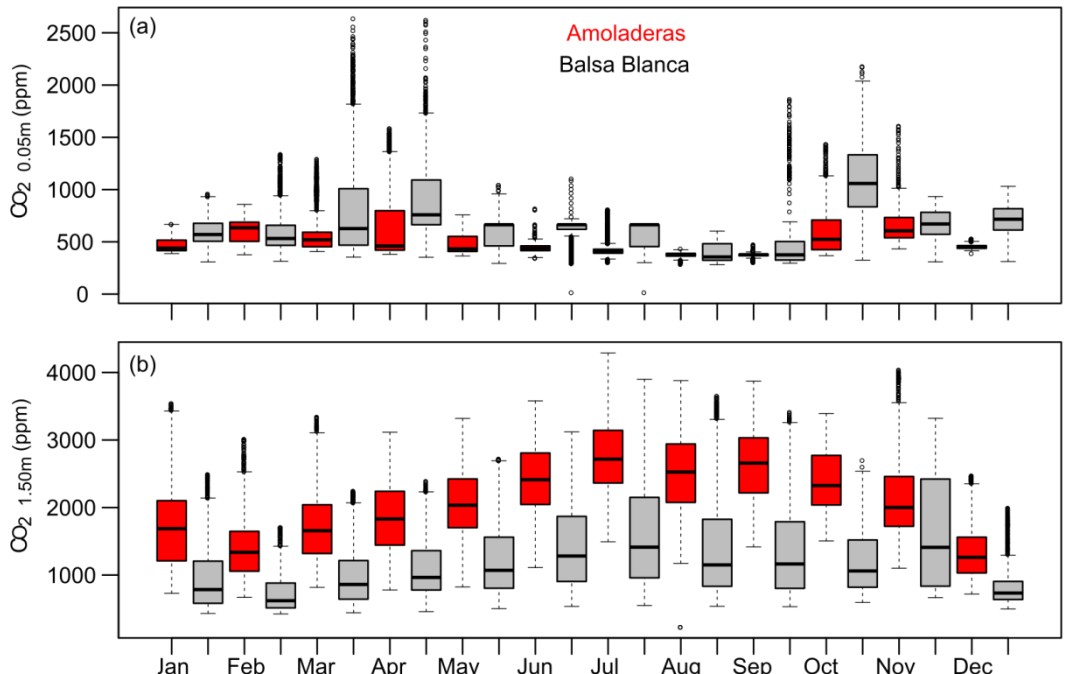

**Figure 8: Box-and-whisker plots of $CO_2$ molar fractions measured at (a) 0.05 m and (b) 1.50 m belowground in Amoladeras (red boxes) and Balsa Blanca (grey boxes) from January 2014 to August 2015. The box extends from the first (Q1) to the third quartiles (Q3) and the central line represents the median (50% percentile). Dots represent**
5 **outliers; upper whisker is located at the smaller of the maximum value and Q3 + 1.5 IQR (Interquartile Range), and lower whisker is located at the larger of the minimum value and Q1 – 1.5 IQR.**

**Table 7: Spearman correlation coefficients ($r_s$) for every paired simple correlation among maximum quality net $CO_2$**
10 **exchange fluxes (µmol m$^{-2}$ s$^{-1}$), absolute and differential pressure (hPa) at 6, 12, 24 and 72hours time-step and absolute and differential $CO_2$ molar fraction measured at 1.50 m below ground (ppm) at the same time-steps. Bold values represent the highest correlation coefficients while shaded ones denotes non-significant relationships (p-values>0.05).**

| | Amoladeras | | | | | | Balsa Blanca | | | | | |
|---|---|---|---|---|---|---|---|---|---|---|---|---|
| | Net $CO_2$ exchange | P | $dP_{6h}$ | $dP_{12h}$ | $dP_{24h}$ | $dP_{72h}$ | Net $CO_2$ exchange | P | $dP_{6h}$ | $dP_{12h}$ | $dP_{24h}$ | $dP_{72h}$ |
| $CO_{2 1.50m}$ | **0.30** | -0.66 | -0.33 | -0.46 | -0.56 | -0.55 | **0.11** | -0.33 | -0.51 | -0.53 | -0.62 | -0.45 |
| $dCO_{2 1.50m\_6h}$ | 0 | -0.1 | -0.84 | -0.5 | -0.26 | -0.01 | 0.02 | -0.07 | -0.63 | -0.19 | -0.13 | -0.04 |
| $dCO_{2 1.50m\_12h}$ | 0.06 | -0.08 | -0.57 | -0.87 | -0.55 | -0.05 | 0.03 | -0.03 | -0.46 | -0.50 | -0.31 | -0.03 |
| $dCO_{2 1.50m\_24h}$ | 0.03 | -0.13 | -0.47 | -0.78 | -0.85 | -0.15 | 0.02 | -0.04 | -0.40 | -0.59 | -0.58 | -0.04 |
| $dCO_{2 1.50m\_72h}$ | 0 | -0.28 | -0.28 | -0.49 | -0.64 | -0.74 | 0.00 | -0.13 | -0.28 | -0.43 | -0.57 | -0.48 |

