# Peer review of "Can land degradation drive differences in the C exchange of two similar semiarid ecosystems?"

_Biogeosciences, 2017_

## Referee Comment (RC1) · Anonymous Referee #1 · 10 Jul 2017

This short paper attempts to describe the impacts of land degradation in semiarid ecosystems on carbon fluxes on the basis of the differences observed between two eddy covariance flux sites in SE Spain. The authors clearly demonstrate that most of the expected meteorological controls over C flux are equivalent between sites, but the carbon fluxes are striking different, varying by a couple of orders of magnitude. As they highlight, this difference in observed net carbon flux is a result of contrasting fluxes of carbon from "subterranean ventilation". As the authors have addressed in other publications, this large carbon efflux cannot be accounted for due to in-situ concurrent biological activity – and this greatly complicates interpretation of contrasting results between the sites, and thus the assessment of the impacts of land degradation.

[Figure]

Unfortunately, the authors do not address this challenge very effectively, and in its current form there is little support for any conclusion about the impacts of land degradation on carbon fluxes. It maybe that the nature of the sites makes it impossible to carry out such a comparison convincingly, but addressing a number of areas is required before this can be determined.

First, the nature of the disturbance and extent to degradation needs to be described in more detail. The similarities between the sites are described in detail, but the crucial differences need more full description than Table 1, and more importantly, the biological implications of this differences (detailed hypotheses) need to be articulated.

Second, these hypotheses need to detail biological controls and the non-biological controls over C fluxes at these two sites, and the fluxes need to be interpreted in that light. In particular, it is differences in productivity that would be key to understanding this. Although it will be difficult given the atypical conditions of a large non-concurrent biological carbon efflux, NEE should be partitioned, and GPP between the sites compared. In addition, there should be a more detailed comparison of the ET fluxes, which in these ecosystems seem to be providing a more comparable indication of ecosystem function. And taken together, it would be interesting to assess inter-site differences in water use efficiency.

Third, the EVI time series as an indicator of productivity requires a closer examination. Given the differences in vegetation cover between the sites (Table 1), it is the similarity in EVI values, rather than the differences (except in the final year), between the two sites that seems most striking. This would suggest that productivity between the sites is not very different, and EVI based GPP estimates would be similar. Does observed tower GPP support this?

Fourth, the downward trend in maximum annual EVI is interesting, and could be investigated more, and potentially over a longer time period. Is it significantly related to a trend in precipitation, and a trend in productivity from the towers? The contrasting

response between the sites in the final year of the record is striking, is it reflected in the tower flux record also – it seems the record is complete over the winter period at least?

Fifth, given that soil $CO_2$ concentration is measured at two depths, is it possible to estimate soil $CO_2$ flux? This could be used to partition the concurrent biological $CO_2$ signal, versus non-biological, and potentially the impacts of degradation on these two different processes.

Overall, a considerable amount of additional analysis is required to separate out the signal from biological and non-biological controls over carbon fluxes from these two sites. It is only then when the flux can be interpreted in terms of vegetation productivity that the impacts of degradation can be assessed in a way that provides insight into processes that are more broadly applicable across semiarid ecosystems.

There are very few grammatical and spelling errors, a few very minor points:

P2 L19 – "concretely" is a strange word choice here and elsewhere – "definitively" is better in some cases, or it can just be removed.

P7 L31 - "punctual' is a strange word choice here – not sure what you are trying to convey

P7 L32 – Daily times series are hard to decipher in Figure 5. Its always a challenge to convey this information. Maybe using a solid black, and ensuring the graphic is a full-page width would help.

P9 L3 – I believe it would be normal to correct pressure to sea-level equivalents before making comparisons such as these.

---

## Referee Comment (RC2) · Anonymous Referee #2 · 16 Jul 2017

The authors compared NEE and biophysical factors between a "natural" grassland site and a "degraded" grassland site in the semiarid area of southeast Spain. They found that the "degraded" site showed less carbon uptake during the growing season but substantially more carbon release during the dry summer months. They attributed the inter-site differences in NEE to higher belowground $CO_2$ concentration at 1.5-m depth and stronger subterranean ventilation at the degraded site. The reported temporal patterns of NEE, ET and EVI at the two sites could promote a better understanding of the effects of land degradation on carbon sequestration in semiarid areas, and provide important information on ecosystem resilience and vulnerability under changing climate. However, I have some concerns regarding how the authors analyzed and interpreted

their data.

Major comments:

1) The authors concluded that "subterranean ventilation of this vadose zone $CO_2$ ... largely drives the differences in C dynamics between them". This conclusion was based on authors' analyses that compare many biophysical factors between the two sites. It turned out that belowground $CO_2$ concentration at 1.5-m depth differed the most between sites. In addition, they found a negative correlation between air pressure and subsoil $CO_2$. However, the reasoning behind this conclusion should be viewed with great caution. Large differences in subsoil $CO_2$ does not necessary explain inter-site differences in NEE. There was no analysis showing a causal link between inter-site variations in subsoil $CO_2$ concentration and $CO_2$ fluxes. Although the authors examined many potential explanatory variables, there still could be other biophysical factors and processes that differ greatly between sites (e.g., soil microbial communities).

2) An unanswered question related to the previous comment is why the degraded site showed such large subsoil $CO_2$ concentrations compared to the natural site. In addition, the degraded site (with much less vegetation cover) showed more carbon release than the natural site. Subterranean ventilation is only a transport process for $CO_2$, but the question is who produced so much $CO_2$? Was it abiotic processes related to carbonate dissolution, or respiratory $CO_2$ production by plants and microbes? This question must be discussed in the paper.

3) The authors only examined NEE dynamics. I would encourage them to also partition NEE and check the two major components of NEE: GPP and Reco. These two components may respond differently to land degradation and interannual climatic variations. Separate analysis on GPP and Reco could provide more information on the differences between the two sites in terms of carbon dynamics.

4) Table 1 showed that the vegetation cover almost three times higher at the natural site than at the degraded site, while the EVI in Figure 5 does not show such a large

difference, at least for most years. So I am wondering whether the pixels you used for extracting EVI well match the location of your ground measurements. Or are there any other reasons for this discrepancy?

5) Writing of the manuscript should still be improved to be more concise and clearer. In addition, there are some very long and complex sentences, which encompass too many ideas (I pointed out some in Specific comments).

Specific comments:

1) Page 1, line 16-20. The background information is a bit too long.

2) Page 1, line 21-24. The sentence is too long and complex. Considering dividing it into shorter sentences.

3) Page 1, line 20. In "global C balance", symbols should be defined upon first mentioning.

4) Page 1, lin3 21. Replace "needs further research" by "still need to be investigated".

5) Page 1, line 25. Please specify what "±" stands for.

6) Page 2, line 1-11. This paragraph is a bit too long. The importance of drylands has been well acknowledged and should only be mentioned very briefly here.

7) Page 2, line 14-22. These case studies are not directly related to this paper. The first sentence of this paragraph already well summarizes the subject of research. I would delete or reduce these case studies.

8) Page 3, line 12-14. I would delete this sentence since the EC technique is a widely used method, and is familiar to most researchers working on carbon exchange.

9) Page 3, line 16. What did you mean by "absorb fast changes"?

10) Page 3, line 25-26. Please specify what kind of "short-term disturbances" you are talking about.

11) Page 3, line 30. Replace "Experimental sites description" with "Site description".

12) Page 4, line 20. The expression "different degradation stages" is not clear to me. More information on the history (degradation, recovery and succession) of the two sites should be provided. A basic question is what caused the degradation?

13) Page 5, line 6. Please specify what "±" stands for.

14) Page 5, line 8. More details on estimating uncertainty should be provided.

15) Page 5, line 15. Please clarify whether or not you took into account above- and below-ground storage terms of heat fluxes when calculating the slopes?

16) Page 5, line 21-22. I would delete this sentence. As you said, it is a widely used index, so there is no need to justify using it.

17) Page 6, line 5-9. Please reword the sentence.

18) Page 6, line 18. Delete "over the study period".

19) Page 6, line 20. The term "annual average precipitation" should be changed to "mean annual precipitation (MAP)".

20) Page 7, line 5. Please delete "(C)" as you have defined it in Introduction.

21) Page 8, line 20-25. I would shorten or remove these sentences since interannual variability is not the focus of this study.

22) Page 9, line 3. Why did you use a threshold value of 1 for Diffst.

23) Page 11, line 25. By saying "stable" did you mean "resilient"?

24) Page 11, line 27. No need to give the definition of "ecosystem resilience". It is a textbook concept that everyone knows.

25) Figure 1. It would be nice if you can add some photos of landscape or vegetation at the two sites.
26) Table 1. The first part of the table (site characteristics) can be removed. These characteristics were well-described in the text and are therefore redundant here. What was the measurement depth for SOC?

27) Figure 3. Adding cumulative GPP and Reco may help understand inter-site differences in carbon dynamics.

28) Table 2. I would put this table in Supplementary Online Materials.

29) Table 6. This table is confusing to me. Can you explain, for example, what does "dCO2, 1.5m_6h" mean?

---

## Author Comment (AC1) · 25 Aug 2017

**Reply to Referee #1**

*We would like to thank referee #1 for the detailed review of our manuscript and the suggestions that will help to improve our manuscript. In the following, we will answer each of the referee's comments.*

This short paper attempts to describe the impacts of land degradation in semiarid ecosystems on carbon fluxes on the basis of the differences observed between two eddy covariance flux sites in SE Spain. The authors clearly demonstrate that most of the expected meteorological controls over C flux are equivalent between sites, but the carbon fluxes are striking different, varying by a couple of orders of magnitude. As they highlight, this difference in observed net carbon flux is a result of contrasting fluxes of carbon from "subterranean ventilation". As the authors have addressed in other publications, this large carbon efflux cannot be accounted for due to in-situ concurrent biological activity – and this greatly complicates interpretation of contrasting results between the sites, and thus the assessment of the impacts of land degradation.

Unfortunately, the authors do not address this challenge very effectively, and in its current form there is little support for any conclusion about the impacts of land degradation on carbon fluxes. It maybe that the nature of the sites makes it impossible to carry out such a comparison convincingly, but addressing a number of areas is required before this can be determined.

**General comments**

1. First, the nature of the disturbance and extent to degradation needs to be described in more detail. The similarities between the sites are described in detail, but the crucial differences need more full description than Table 1, and more importantly, the biological implications of these differences (detailed hypotheses) need to be articulated.

*We agree with the reviewer. Therefore, the following paragraph will be added to the revised manuscript in page 3 line 20:*

*"Some land degradation processes are evident when we compare the "natural" site with the "degraded" site. This land degradation processes can directly affect abiotic and/or biotic factors, which in turn influence the biological*

*and/or non-biological processes that compose the net ecosystem $CO_2$ exchange. Firstly, vegetation cover is almost 3 times higher in Balsa Blanca (BB), the "natural" site; this implies for the degraded site higher thermal and radiative stress in the soil, especially during the drought period (Rey et al., 2017). The overall hypothesized effects of this degradation driver on biological processes are a direct reduction in plant productivity and respiration, and an indirect decrease in heterotrophic respiration. Secondly, the higher cover of bare soil and outcrops in Amoladeras (AMO), the "degraded" site, may increase the soil-atmosphere interconnectivity, which indirectly can enhance advective $CO_2$ release through subterranean ventilation. Thirdly, the reduced soil fertility and depth may provoke changes in microbial communities (Evans and Wallenstein, 2014) due to stronger nutrient and water limitations. Consequently, a direct decrease in heterotrophic respiration and plant productivity and respiration is expected."*

2. Second, these hypotheses need to detail biological controls and the non-biological controls over C fluxes at these two sites, and the fluxes need to be interpreted in that light. In particular, it is differences in productivity that would be key to understanding this. Although it will be difficult given the atypical conditions of a large non-concurrent biological carbon efflux, NEE should be partitioned, and GPP between the sites compared. In addition, there should be a more detailed comparison of the ET fluxes, which in these ecosystems seem to be providing a more comparable indication of ecosystem function. And taken together, it would be interesting to assess inter-site differences in water use efficiency.

*According to the referee's suggestion, we have partitioned the net $CO_2$ fluxes ($F_c$) in order to assess the potential direct influence of land degradation on the gross primary production (GPP) and ecosystem respiration ($R_{eco}$) components. Given the extreme $CO_2$ release detected due to subterranean ventilation, two steps have been performed at each site.*

*Firstly, we have modelled the ventilative $CO_2$ efflux by adapting the approach proposed by (Pérez-Priego et al., 2013) with the results of previous studies performed in both sites (López-Ballesteros et al., 2016; 2017). Essentially, we aimed to isolate those moments when subterranean ventilation ($V_n$) dominates the $F_c$ and biological fluxes are negligible. These moments correspond to daytime hours during the extremely dry periods. Data were selected using the following conditions:*

    *(i)      Net radiation $> 10\ W\ m^{-2}$*
    *(ii)    8 < Daily averaged bowen ratio < 10*
    *(iii)   Daily soil water content (in bare soil) $< 10^{th}$ percentile (Amoladeras)*
           *and $< 20^{th}$ percentile (Balsa Blanca)*

*A less restrictive threshold was used in Balsa Blanca in order to get enough data to build the $V_n$ model, since long-term data gaps occurred in Balsa Blanca during the summer seasons of 2012, 2014 and 2015. Afterwards, in order to build the linear model of $V_n$, these selected $F_c$ data (maximum quality; QC flag=0) were related to the friction velocity ($u_*$). The model results for both sites are shown below:*

[Figure]

*Figure S1: Half-hourly net $CO_2$ fluxes of maximum quality (QC flag=0) versus friction velocity ($u_*$) corresponding to daytime hours during the extremely dry periods when subterranean ventilation dominates the net $CO_2$ flux. Red and black dots represent Amoladeras and Balsa Blanca, respectively.*

| Model parameters | Amoladeras | Balsa Blanca |
|---|---|---|
| *Intercept ± error (p-value)* | *-1.876 ± 0.291 (4e-09)* | *0.628 ± 0.508 (0.226)* |
| *Slope ± error (p-value)* | *8.500 ± 0.549 (<2e-16)* | *0.578 ± 0.944 (0.545)* |
| $R^2$ | *0.706 (<2.2e-16)* | *0.013 (0.5451)* |
| *n* | *102* | *31* |

*Table 3: Linear regression results between half-hourly net $CO_2$ fluxes of maximum quality (QC flag=0) and friction velocity ($u_*$) used to model subterranean ventilation.*

*As the table above shows, the $V_n$ model is uniquely valid for Amoladeras.*

*Therefore, we only applied the $V_n$ model to Amoladeras data, concretely, during those periods were ventilation occurs according to previous research (López-Ballesteros et al., 2017):*

*(i)      Net radiation > 10 W m$^{-2}$*
*(ii)     Daily averaged bowen ratio > 4*
*(iii)    Daily soil water content (in bare soil) < 0.01 m$^3$ m$^{-3}$*
*(iv)     $\sigma_{swc}$ (daily variance of soil water content in bare soil)< 5e-6 (m$^3$ m$^{-3}$)$^2$*

*We use those moments with very low $\sigma_{swc}$ in order to discern $R_{eco}$ increases caused by rain pulses (Birch effect) from $V_n$ fluxes during the dry season.*

*Then, the modelled ventilative fluxes were substracted from the measured $F_c$ to obtain the $F_c$ corresponding only to biological processes (i.e. biological $F_c$; see Figure below).*

[Figure]

*Figure S1: Cumulative measured and biological (after applying the ventilation model) net CO$_2$ exchange for every month of the study period (5 hydrological years; 2009-2015) in Amoladeras.*

*Secondly, the partitioning approach proposed by Lasslop et al. (2010) was applied to the biological $F_c$ for both sites in order to obtain GPP and $R_{eco}$ fluxes. We chose this approach given the determinant influence of hydric stress, in this case*

*atmospheric drought (assessed via VPD), on the physiology of Machrocloa tenacissima, the dominant plant species of the studied semiarid ecosystems (Pugnaire et al., 1996; López-Ballesteros et al. 2016).*

*This information would be added to a new section within the manuscript, concretely the subsection "2.3. Flux partitioning to estimate GPP and $R_{eco}$" within the Material and Methods section.*

[Figure]

*Figure 7: Monthly cumulative fluxes of (a) biological net ecosystem $CO_2$ exchange, (b) ecosystem respiration ($R_{eco}$), (c) negative gross primary production and (d) water use efficiency over the six hydrological years of study (2009-2015) for Amoladeras (dark red) and Balsa Blanca (grey). Lacking bars correspond to long-term data losses.*

Finally, the results of the "biological" annual carbon balance are in accordance with the hypotheses, since annual C emission was always measured at the "degraded" site, whereas the "natural" site acted as a neutral and mild C sink. On average, Amoladeras emitted 32 g C m$^{-2}$ more than Balsa Blanca.

| Year | Amoladeras | Balsa Blanca |
|---|---|---|
| 2009/2010 | 3+-7 | -32+-10 |
| 2012/2013 | 28+-5 | 0+-8 |

We could not compare the annual C balance of 2010/2011 between sites due to a long-term data gap in the u$_*$ time series in Amoladeras during the spring months (February-April).

During autumn, monthly biological net $CO_2$ fluxes were, on average, ~4 times higher at the "natural" site, excepting the last study year, when the net $CO_2$ emission at the "degraded" site was 21 times greater than at the "natural" site. However, during winter and spring months, net $CO_2$ uptake was generally higher at the "natural" site (Fig. 5a).

On average, during the six years of study, GPP, $R_{eco}$ and WUE were nine, twice and ten times higher, respectively, at the "natural" site compared to the "degraded" site. Firstly, GPP was always higher at the "natural" site compared to the "degraded" site (Fig. 7c). Major differences occurred in autumn 2014/2015, when monthly cumulative GPP at the "natural" site was 32 times higher on average. Similarly, $R_{eco}$ was generally higher, up to 786% (October 2014), at the "natural" site. However, respiratory fluxes were occasionally greater at the "degraded" site, from 2% to 31% higher, during spring and winter months of all studied years excepting 2013/2014 (Fig .7b). Maximum inter-site differences in GPP and $R_{eco}$ were found in winter and autumn 2014/2015, following the driest year, when monthly GPP was, on average, ~30 times higher at the "natural" site compared with the "degraded" site. Similarly, monthly $R_{eco}$ was ~5 times greater at the "natural" site. Inter-site differences in partitioned fluxes could not be assessed during spring months due to the lack of data from the "natural site". Secondly, Water Use Efficiency (WUE) was lower at "the "degraded" site showed during the whole study period, when maximum and minimum differences coincided with the highest and lowest differences in GPP between sites. On average, monthly WUE was 6 and 1.5 times higher in the "natural" site during winter and spring. Major inter-site differences were found in autumn and winter 2014/2015 (Fig. 7d).

*This information would be added to a new section within the manuscript, the subsection "3.4. Biological Net Ecosystem Exchange, Gross Primary Production, Ecosystem Respiration and Water Use Efficiency" within the Results section. Accordingly, these results would be discussed and related to the study hypotheses in the Discussion section.*

[Figure]

*Figure 5: Cumulative seasonal evapotranspiration fluxes (ET) over the study period in both experimental sites. In case of Balsa Blanca, lacking bars correspond to long-term data losses (>50% data). Error bars denote uncertainty derived from the gap-filling procedure.*

*Apart from that, ET results showed ~30% higher ET at the "natural" site compared to the "degraded" site during spring. Major inter-site differences in autumn occurred in the first and last year of study, when ET was 23% and 12% higher at BB, respectively. In this regard, we think that the "natural" site shows more capacity to maintain water availability during the growing season, however, the*

*lack of data complicates the interpretation. The higher uncertainty of ET data in Amoladeras is due to a higher fraction of short-term data gaps compared to Balsa Blanca, on average annual fraction of data losses is 27% higher in Amoladeras.*

*This information would be added to the subsection 3.3., which would be renamed as "Seasonal and diurnal net $CO_2$ and water vapor exchanges" within the Results section. Accordingly, these results will be discussed in the Discussion section.*

3. Third, the EVI time series as an indicator of productivity requires a closer examination. Given the differences in vegetation cover between the sites (Table 1), it is the similarity in EVI values, rather than the differences (except in the final year), between the two sites that seems most striking. This would suggest that productivity between the sites is not very different, and EVI based GPP estimates would be similar. Does observed tower GPP support this?

*As stated before, we have found that GPP was always higher at the "natural" site compared to the "degraded" site. Thus, there is a discrepancy between GPP estimates and EVI values. We think that this is due to the different spatial scales defining every measurement. MODIS pixels have an area of ~6.25 ha while the eddy covariance footprint corresponds to a smaller area of ~1ha. Therefore, there is an EVI uncertainty that stems from the influence of other surface elements apart from vegetation, such as bare soil or outcrops within the pixel, which is our case. In fact, previous studies confirm the discrepancy between MODIS- and EC-derived GPP estimates, especially on sparse vegetation areas with low productivity (Gilabert et al., 2015).*

*This information would be added to the fifth paragraph of the Discussion section of the revised manuscript.*

4. Fourth, the downward trend in maximum annual EVI is interesting, and could be investigated more, and potentially over a longer time period. Is it significantly related to a trend in precipitation, and a trend in productivity from the towers? The contrasting response between the sites in the final year of the record is striking, is it reflected in the tower flux record also – it seems the record is complete over the winter period at least?

*The period we are studying is too short to assess trends. In addition, precipitation in this region is quite variable and as said before, the direct comparison between EVI and GPP estimates from EC measurements can lead to biased conclusions. Apart*

*from that, if we look to a wider time window, by using longer time series of EVI and precipitation, we can realize that the precipitation in 2009/2010 was extremely high compared with the annual precipitation of the experimental sites, which equates to 220 mm (Table 1). Thus, we believe that instead of a decreasing trend what we see is a pulse response following the wettest year, as can be seen in the figures below. In fact, inter-site differences in EVI are greater before the study period, from 2000 to 2009, compared to 2009-2014. During the last year of the study period, this difference became similar to the pattern observed before the wettest year.*

[Figure]

*Notice that precipitation data shown in the figure above have not been measured in the experimental sites but quite near them since the EC stations were installed in 2006 (Balsa Blanca) and 2007 (Amoladeras). Concretely, the agro-climatic stations, "Almería" and "Níjar", where precipitation was measured, are 13.55 km and 11.22 km from Amoladeras and Balsa Blanca, respectively.*

*Source:*
https://www.juntadeandalucia.es/agriculturaypesca/ifapa/ria/servlet/FrontController?action=Init

5. Fifth, given that soil $CO_2$ concentration is measured at two depths, is it possible to estimate soil $CO_2$ flux? This could be used to partition the concurrent biological $CO_2$ signal, versus non-biological, and potentially the impacts of degradation on these two different processes.

*The estimation of soil $CO_2$ efflux by using the gradient method (Sánchez-Cañete and Kowalski, 2014) assumes that the release of $CO_2$ from soil is performed by diffusion exclusively. However, as demonstrated in previous studies, advection (non-diffusive transport) can play an important role in the soil-atmosphere gaseous exchange (Kowalski et al., 2008; Sanchez-Cañete et al., 2011; Sánchez-Cañete et al., 2016; López-Ballesteros et al., 2017; Serrano-Ortiz et al., 2009; Subke et al., 2003; Risk et al., 2013; Roland et al., 2015). Further isotopic analyses are necessary to assess the role of biological vs non-biological $CO_2$ production processes in soil $CO_2$ efflux as well as to determine the transport processes driving the soil-atmosphere net $CO_2$ exchange. Unfortunately, we do not have these results, although we plan to work on it in the near future. Additionally, through the application of the ventilation model (previously explained), we have discriminated between the biological and non-biological net ecosystem $CO_2$ exchange in Amoladeras.*

Overall, a considerable amount of additional analysis is required to separate out the signal from biological and non-biological controls over carbon fluxes from these two sites. It is only then when the flux can be interpreted in terms of vegetation

productivity that the impacts of degradation can be assessed in a way that provides insight into processes that are more broadly applicable across semiarid ecosystems.

*We believe that the empirical ventilation model that we have added (see above) should satisfy the referee in this regard.*

**Specific comments**

There are very few grammatical and spelling errors, a few very minor points:

P2 L19 – "concretely" is a strange word choice here and elsewhere – "definitively" is better in some cases, or it can just be removed.

*We agree and we have removed "concretely" from the sentence.*

P7 L31 - "punctual' is a strange word choice here – not sure what you are trying to convey.

*We agree about "punctual" should be removed from the sentence. What we wanted to say is that EVI data is discrete as oppose to fluxes but this information is already explained in the material and methods section.*

P7 L32 – Daily time series are hard to decipher in Figure 5. It is always a challenge to convey this information. Maybe using a solid black, and ensuring the graphic is a full-page width would help.

*We believe that this picture carries much information that has to be shown and complements other figures (Fig. 4, 5 and 7) where flux data are aggregated. After trying several graphical options, we decided to use dashed lines because solid black lines (Balsa Blanca data) mask the red lines (Amoladeras data). We have increased the width of the figure and thickened the black lines.*

P9 L3 – I believe it would be normal to correct pressure to sea-level equivalents before making comparisons such as these.

*We agree so we have corrected pressure to sea level equivalents using the hypsometric equation and afterwards we have computed the Wilcoxon test for the different analysis periods. Results show a smaller difference between sites in corrected pressure compared to uncorrected values. These results (table below) would substitute previous ones (Tables 4, S1, S2 and S3) in the revised version of the manuscript.*

| Period | Diff | $Diff_{st}$ | p-value | n |
|---|---|---|---|---|
| All periods | 2.3226 | 0.3737 | 0 | 166336 |
| May-September | 2.2120 | 0.5828 | 0 | 71188 |
| May-September Daytime | 2.1101 | 0.5602 | 0 | 34280 |

Evans, S. E., and Wallenstein, M. D.: Climate change alters ecological strategies of soil bacteria, Ecology Letters, 17, 155-164, 10.1111/ele.12206, 2014.

Gilabert, M. A., Moreno, A., Maselli, F., Martínez, B., Chiesi, M., Sánchez-Ruiz, S., García-Haro, F. J., Pérez-Hoyos, A., Campos-Taberner, M., Pérez-Priego, O., Serrano-Ortiz, P., and Carrara, A.: Daily GPP estimates in Mediterranean ecosystems by combining remote sensing and meteorological data, ISPRS Journal of Photogrammetry and Remote Sensing, 102, 184-197, https://doi.org/10.1016/j.isprsjprs.2015.01.017, 2015.

Kowalski, A. S., Serrano-Ortiz, P., Janssens, I. A., Sánchez-Moral, S., Cuezva, S., Domingo, F., Were, A., and Alados-Arboledas, L.: Can flux tower research neglect geochemical CO2 exchange?, Agricultural and Forest Meteorology, 148, 1045-1054, 2008.

Lasslop, G., Reichstein, M., Papale, D., Richardson, A. D., Arneth, A., Barr, A., Stoy, P., and Wohlfahrt, G.: Separation of net ecosystem exchange into assimilation and respiration using a light response curve approach: critical issues and global evaluation, Global Change Biology, 16, 187-208, 10.1111/j.1365-2486.2009.02041.x, 2010.

López-Ballesteros, A., Serrano-Ortiz, P., Kowalski, A. S., Sánchez-Cañete, E. P., Scott, R. L., and Domingo, F.: Subterranean ventilation of allochthonous CO2 governs net CO2 exchange in a semiarid Mediterranean grassland, Agricultural and Forest Meteorology, 234–235, 115-126, http://dx.doi.org/10.1016/j.agrformet.2016.12.021, 2017.

Pérez-Priego, O., Serrano-Ortiz, P., Sánchez-Cañete, E. P., Domingo, F., and Kowalski, A. S.: Isolating the effect of subterranean ventilation on CO2 emissions from drylands to the atmosphere, Agricultural and Forest Meteorology, 180, 194-202, 2013.

Pugnaire, F. I., Haase, P., Incoll, L. D., and Clark, S. C.: Response of the tussock grass Stipa tenacissima to watering in a semi-arid environment, Functional Ecology, 10, 265-274, 1996.

Rey, A., Oyonarte, C., Morán-López, T., Raimundo, J., and Pegoraro, E.: Changes in soil moisture predict soil carbon losses upon rewetting in a perennial semiarid steppe in SE Spain, Geoderma, 287, 135-146, http://dx.doi.org/10.1016/j.geoderma.2016.06.025, 2017.

Risk, D., Lee, C. K., MacIntyre, C., and Cary, S. C.: First year-round record of Antarctic Dry Valley soil CO2 flux, Soil Biology and Biochemistry, 66, 193-196, http://dx.doi.org/10.1016/j.soilbio.2013.07.006, 2013.

Roland, M., Vicca, S., Bahn, M., Ladreiter-Knauss, T., Schmitt, M., and Janssens, I. A.: Importance of nondiffusive transport for soil CO2 efflux in a temperate mountain grassland, Journal of Geophysical Research: Biogeosciences, 120, 502-512, 10.1002/2014jg002788, 2015.

Sanchez-Cañete, E. P., Serrano-Ortiz, P., Kowalski, A. S., Oyonarte, C., and Domingo, F.: Subterranean CO2 ventilation and its role in the net ecosystem carbon balance of a karstic shrubland, Geophysical Research Letters, 38, 2011.

Sánchez-Cañete, E. P., and Kowalski, A. S.: Comment on "Using the gradient method to determine soil gas flux: A review" by M. Maier and H. Schack-Kirchner, Agricultural and Forest Meteorology, 197, 254-255, http://dx.doi.org/10.1016/j.agrformet.2014.07.002, 2014.

Sánchez-Cañete, E. P., Oyonarte, C., Serrano-Ortiz, P., Curiel Yuste, J., Pérez-Priego, O., Domingo, F., and Kowalski, A. S.: Winds induce CO2 exchange with the atmosphere and vadose zone transport in a karstic ecosystem, Journal of Geophysical Research: Biogeosciences, n/a-n/a, 10.1002/2016jg003500, 2016.

Serrano-Ortiz, P., Domingo, F., Cazorla, A., Were, A., Cuezva, S., Villagarcía, L., Alados-Arboledas, L., and Kowalski, A. S.: Interannual CO2 exchange of a sparse Mediterranean shrubland on a carbonaceous substrate, Journal of Geophysical Research G: Biogeosciences, 114, 2009.

Subke, J.-A., Reichstein, M., and Tenhunen, J. D.: Explaining temporal variation in soil CO2 efflux in a mature spruce forest in Southern Germany, Soil Biology and Biochemistry, 35, 1467-1483, http://dx.doi.org/10.1016/S0038-0717(03)00241-4, 2003.

---

## Author Comment (AC2) · 25 Aug 2017

**Reply to Referee #2**

*We would like to thank referee #2 for the detailed review of our manuscript and the suggestions that have helped to improve our manuscript. In the following, we will answer each of the referee's comments.*

The authors compared NEE and biophysical factors between a "natural" grassland site and a "degraded" grassland site in the semiarid area of southeast Spain. They found that the "degraded" site showed less carbon uptake during the growing season but substantially more carbon release during the dry summer months. They attributed the inter-site differences in NEE to higher belowground CO2 concentration at 1.5-m depth and stronger subterranean ventilation at the degraded site. The reported temporal patterns of NEE, ET and EVI at the two sites could promote a better understanding of the effects of land degradation on carbon sequestration in semiarid areas, and provide important information on ecosystem resilience and vulnerability under changing climate. However, I have some concerns regarding how the authors analyzed and interpreted their data.

**General comments**

1. The authors concluded that "subterranean ventilation of this vadose zone CO2 largely drives the differences in C dynamics between them". This conclusion as based on authors' analyses that compare many biophysical factors between the two sites. It turned out that belowground CO2 concentration at 1.5-m depth differed the most between sites. In addition, they found a negative correlation between air pressure and subsoil CO2. However, the reasoning behind this conclusion should be viewed with great caution. Large differences in subsoil CO2 does not necessary explain inter-site differences in NEE. There was no analysis showing a causal link between inter-site variations in subsoil CO2 concentration and CO2 fluxes. Although the authors examined many potential explanatory variables, there still could be other biophysical factors and processes that differ greatly between sites (e.g., soil microbial communities).

*Although we did not test the direct link between subsoil $CO_2$ molar fraction at 1.5m and net $CO_2$ exchange fluxes, our results lead to this hypothesis, since inter-site differences in meteorological variables are minimal. On the other hand, although measurements of the metabolic activity of microbial communities would help to understand inter-site differences in heterotrophic respiration patterns, we think that*

*a potential difference in soil microbial communities should not totally explain the important carbon release observed in the "degraded" site during the dry season, mainly due to a lack of water in the soil at both sites. Apart from that, a previous study performed at Amoladeras relates this carbon release to atmospheric turbulence through linear regressions with friction velocity ($u_*$), and also shows an influence of net radiation and VPD in the $CO_2$ fluxes (López-Ballesteros et al., 2017). Overall, based on other investigations performed by this research group in several ecosystems located in the same province (Kowalski et al., 2008;Sanchez-Cañete et al., 2011;Serrano-Ortiz et al., 2009;Sánchez-Cañete et al., 2013;Pérez-Priego et al., 2013), we strongly believe that there is an advective transport of stored $CO_2$-rich air from the vadose zone to the atmosphere, especially under high hydric stress and high turbulence conditions.*

*Apart from this, we have included in Table 4, the linear regression results for subsoil $CO_2$ at 1.5m and net $CO_2$ exchange fluxes, and as expected, a higher correlation was obtained in Amoladeras.*

***Table 4:*** *Spearman correlation coefficients ($r_s$) for every paired simple correlation among maximum quality net $CO_2$ exchange fluxes (µmol m$^{-2}$ s$^{-1}$), absolute and differential pressure (hPa) at 6, 12, 24 and 72hours time-step and absolute and differential $CO_2$ molar fraction measured at 1.50 m below ground (ppm) at the same time-steps. Bold values represent the highest correlation coefficients while shaded ones denotes non-significant relationships (p-values>0.05).*

| | Amoladeras | | | | | | Balsa Blanca | | | | | |
|---|---|---|---|---|---|---|---|---|---|---|---|---|
| | Net $CO_2$exchange | P | $dP_{6h}$ | $dP_{12h}$ | $dP_{24h}$ | $dP_{72h}$ | Net $CO_2$exchange | P | $dP_{6h}$ | $dP_{12h}$ | $dP_{24h}$ | $dP_{72h}$ |
| $CO_{21.50m}$ | **0.30** | -0.66 | -0.33 | -0.46 | -0.56 | -0.55 | **0.11** | -0.33 | -0.51 | -0.53 | -0.62 | -0.45 |
| $dCO_{21.50m\_6h}$ | 0 | -0.1 | -0.84 | -0.5 | -0.26 | -0.01 | 0.02 | -0.07 | -0.63 | -0.19 | -0.13 | -0.04 |
| $dCO_{21.50m\_12h}$ | 0.06 | -0.08 | -0.57 | -0.87 | -0.55 | -0.05 | 0.03 | -0.03 | -0.46 | -0.50 | -0.31 | -0.03 |
| $dCO_{21.50m\_24h}$ | 0.03 | -0.13 | -0.47 | -0.78 | -0.85 | -0.15 | 0.02 | -0.04 | -0.40 | -0.59 | -0.58 | -0.04 |
| $dCO_{21.50m\_72h}$ | 0 | -0.28 | -0.28 | -0.49 | -0.64 | -0.74 | 0.00 | -0.13 | -0.28 | -0.43 | -0.57 | -0.48 |

2. An unanswered question related to the previous comment is why the degraded site showed such large subsoil CO2 concentrations compared to the natural site. In addition, the degraded site (with much less vegetation cover) showed more carbon release than the natural site. Subterranean ventilation is only a transport process for

CO2, but the question is who produced so much CO2? Was it abiotic processes related to carbonate dissolution, or respiratory CO2 production by plants and microbes? This question must be discussed in the paper.

*As explained by López-Ballesteros et al. (2017), the potential origins of the released $CO_2$ could be geological degassing and/or subterranean translocation of $CO_2$ in both gaseous and aqueous phases. In this publication, there is a detailed argument of these two hypotheses and also it is recognized that future research is needed in order to understand how $CO_2$ transport and production processes interact and modulate drylands' terrestrial C balance.*

3. The authors only examined NEE dynamics. I would encourage them to also partition NEE and check the two major components of NEE: GPP and Reco. These two components may respond differently to land degradation and interannual climatic variations. Separate analysis on GPP and Reco could provide more information on the differences between the two sites in terms of carbon dynamics.

*In accordance to the comment of the referee #1 and your suggestion, we have performed the flux partitioning proposed by Lasslop et al.(2010) in order to estimate the magnitude of GPP and $R_{eco}$ for both sites. However, in the case of Amoladeras, we first subtracted the flux magnitude corresponding to subterranean ventilation by applying a ventilation model (Pérez-Priego et al., 2013). The methodology used as well as the results are explained in the reply to referee #1, and would be included in the revised version of the manuscript.*

4. Table 1 showed that the vegetation cover almost three times higher at the natural site than at the degraded site, while the EVI in Figure 5 does not show such a large difference, at least for most years. So I am wondering whether the pixels you used for extracting EVI well match the location of your ground measurements. Or are there any other reasons for this discrepancy?

*We have verified that the chosen pixels match the location of our ground measurements. Since referee #1 made the same remark, please read the answer to the fourth general comment of the reply to referee #1.*

**Specific comments**

1. Page 1, line 16-20. The background information is a bit too long.

*We have modified these sentences as follows: "Currently, drylands occupy more than one third of the global terrestrial surface and are recognized as areas vulnerable to land degradation. The concept of land degradation stems from the loss*

*of an ecosystem's biological productivity, due to long-term loss of natural vegetation or depletion of soil nutrients."*

2. Page 1, line 21-24. The sentence is too long and complex. Considering dividing it into shorter sentences.

*We have rewritten this sentence as follows: "In the present study, we compare net carbon C and water vapor fluxes, together with meteorological and satellite data and vadose zone measurements ($CO_2$, water content and temperature) between two nearby (~23 km) experimental sites representing "natural" (i.e. site of reference) and "degraded" grazed semiarid grasslands. We utilized data acquired in two eddy covariance stations located in SE Spain during 6 years with highly variable precipitation magnitude and distribution."*

3. Page 1, line 20. In "global C balance", symbols should be defined upon first mentioning.

*We have defined the symbol at this sentence.*

4. Page 1, lin3 21. Replace "needs further research" by "still need to be investigated".

*We have followed your suggestion.*

5. Page 1, line 25. Please specify what "_" stands for.

*We did not find that character in the sentence.*

6. Page 2, line 1-11. This paragraph is a bit too long. The importance of drylands has been well acknowledged and should only be mentioned very briefly here.

*We have shortened this paragraph as follows: "The concept of land degradation stems from the loss of an ecosystem's biological productivity, which in turn relies on several degradation processes such as long-term loss of natural vegetation, deterioration of soil quality, depletion in biodiversity or water and wind erosion (UNCCD, 1994). Drylands (arid, semiarid and dry sub-humid areas), which occupy more than one third of Earth´s land surface and are inhabited by more than 2 billion people (Niemeijer et al., 2005), have been recognized as areas vulnerable to land degradation processes. In fact, they have expanded globally for the last sixty years at an estimated annual rate of 5.8 million hectares in mid latitudes alone (Lal, 2001), and are projected to expand under future climate change scenarios (Feng and Fu, 2013; Cook et al., 2014), especially in the Mediterranean region, where major expansions of semiarid areas will occur (Gao and Giorgi, 2008; Feng and Fu, 2013)."*

7. Page 2, line 14-22. These case studies are not directly related to this paper. The first sentence of this paragraph already well summarizes the subject of research. I would delete or reduce these case studies.

*We have reduced the case studies as suggested. We want to mention regions or countries where desertification has been assessed previously in order to assess the spatial representativeness of land degradation research globally. However, we have followed your suggestion by deleting the last sentence of this paragraph which talks about the global studies using modelling approaches, since our study is local.*

8. Page 3, line 12-14. I would delete this sentence since the EC technique is a widely used method, and is familiar to most researchers working on carbon exchange.

*We agree and have deleted that sentence.*

9. Page 3, line 16. What did you mean by "absorb fast changes"?

*Here, we are talking about resilience, hence we have rewritten this sentence as follows: "Owing to the high temporal resolution of the EC method, we can assess the effect of land degradation as a slow change or disturbance legacy in the studied ecosystems and how, in turn, it influences ecosystem resilience to short-term disturbances, such as climate extremes (i.e. droughts, heat waves)."*

10. Page 3, line 25-26. Please specify what kind of "short-term disturbances" you are talking about.

*As written before, we meant climate extremes, such as droughts or heat waves.*

11. Page 3, line 30. Replace "Experimental sites description" with "Site description".

*Done.*

12. Page 4, line 20. The expression "different degradation stages" is not clear to me. More information on the history (degradation, recovery and succession) of the two sites should be provided. A basic question is what caused the degradation?

*The stronger degradation effects observed in Amoladeras ("degraded" site) compared to Balsa Blanca ("natural" site) are probably due to its proximity to populated areas. The main factor provoking degradation in this Mediterranean area was the increase of rural population from the beginning of the 20th century until late 1950s (Grove and Rackham, 2001). At that time, timber extraction, the use of tussock fiber for textile manufacturing and extensive farming were common economic activities potentially causing a higher anthropic pressure on the*

*"degraded" site. Afterwards, rural exodus during the mid-century involved the abandonment of this agriculture and farming practices. However, although degradation drivers are not currently active, their effects are still observable in the area; this is a case of "relict" degradation (Puigdefábregas and Mendizábal, 2004).*

*We will add this information in the revised version of the manuscript.*

13. Page 5, line 6. Please specify what "" stands for.

*We did not find that character in the sentence. It seems that the referee has a problem with his/her PDF viewer, particularly regarding the symbol "±".*

14. Page 5, line 8. More details on estimating uncertainty should be provided.

*We have rewritten this paragraph as follows: "Missing data were gap-filled by means of the marginal distribution approach proposed by Reichstein et al. (2005) and uncertainty derived from the gap-filling procedure by using the variance of the measured data, which was calculated by introducing artificial gaps and repeating the standard gap-filling procedure. Twice the standard deviation of sums of total data was taken as the uncertainty for the several aggregating time periods we used in the analysis."*

15. Page 5, line 15. Please clarify whether or not you took into account above- and below-ground storage terms of heat fluxes when calculating the slopes?

*We have included the following sentence to clarify it: "The storage term in the soil heat fluxes was included in the estimates while in case of sensible and latent heat fluxes, this term was negligible given the short height of the vegetation (~50 cm)."*

16. Page 5, line 21-22. I would delete this sentence. As you said, it is a widely used index, so there is no need to justify using it.

*Done.*

17. Page 6, line 5-9. Please reword the sentence.

*We have reworded these sentences as follows: "This test was chosen because the variables used satisfied the independence and continuity assumptions but not all were normally distributed. The confidence level used was 95%. The effect size was evaluated using the median of the difference between the samples (Amoladeras minus Balsa Blanca), which was expressed as a standardized value (divided by its standard deviation; $Diff_{st}$; dimensionless) in order to be able to compare results among different variables."*

18. Page 6, line 18. Delete "over the study period".

*Done.*

19. Page 6, line 20. The term "annual average precipitation" should be changed to "mean annual precipitation (MAP)".

*Done.*

20. Page 7, line 5. Please delete "(C)" as you have defined it in Introduction.

*Done.*

21. Page 8, line 20-25. I would shorten or remove these sentences since interannual variability is not the focus of this study.

*We have shortened this paragraph as follows: "On the other hand, differences in the inter-annual variability of EVI were found between years. Concretely, 2009/2010 and 2013/2014 were the years with maximum and minimum annual precipitation and EVI observations, respectively, for both sites. In 2009/2010, EVI observations were 28% and 20% higher than the six-year averaged values in BB and AMO, respectively. In case of the driest year, 2013/2014, the growing season (winter-spring) EVI was reduced 35% and 28% in BB and AMO, respectively."*

22. Page 9, line 3. Why did you use a threshold value of 1 for Diffst.

*We chose that threshold just to comment the data, all results are shown in Tables 4 and 5, and therefore they can be interpreted by the readers.*

23. Page 11, line 25. By saying "stable" did you mean "resilient"?

*Yes. We have substituted "stable" with "resilient".*

24. Page 11, line 27. No need to give the definition of "ecosystem resilience". It is a textbook concept that everyone knows.

*We have removed the definition of "ecosystem resilience".*

25. Figure 1. It would be nice if you can add some photos of landscape or vegetation at the two sites.

*We agree and we would include the picture below.*

[Figure]

[Figure]

26. Table 1. The first part of the table (site characteristics) can be removed. These characteristics were well-described in the text and are therefore redundant here. What was the measurement depth for SOC?

*We agree so we have deleted the first part of Table 1. Regarding the SOC measurement, given that these soils are shallow with maximum depth of 20 cm, we only took a composite sample of the profile.*

27. Figure 3. Adding cumulative GPP and Reco may help understand inter-site differences in carbon dynamics.

*Given the lack of flux data in both sites, instead of adding a figure of cumulative GPP and $R_{eco}$, we have added the following figure showing the monthly cumulative NEE (biological), GPP , $R_{eco}$ and Water Use Efficiency (WUE) for both sites over the whole study period.*

[Figure]

*Figure 7: Monthly cumulative fluxes of (a) biological net ecosystem $CO_2$ exchange, (b) ecosystem respiration ($R_{eco}$), (c) negative gross primary production and (d) water use efficiency over the six hydrological years of study (2009-2015) for Amoladeras (dark red) and Balsa Blanca (grey). Lacking bars correspond to long-term data losses.*

28. Table 2. I would put this table in Supplementary Online Materials.

*We believe that this information is crucial to interpret and compare these results with other studies. Hence, we do not think that it should be placed in the Supplementary Online Materials.*

29. Table 6. This table is confusing to me. Can you explain, for example, what does "dCO2, 1.5m_6h" mean?

*dCO2, 1.5m_6h means the difference between the $CO_2$ molar fraction measured at a time "t" and the $CO_2$ molar fraction measured 6 hours before "t-6h". This analysis was performed to show how different is the influence of pressure variations in the subsoil $CO_2$ molar fraction between sites, as demonstrated at another study in the area (Sánchez-Cañete et al., 2013).*

Gilabert, M. A., Moreno, A., Maselli, F., Martínez, B., Chiesi, M., Sánchez-Ruiz, S., García-Haro, F. J., Pérez-Hoyos, A., Campos-Taberner, M., Pérez-Priego, O., Serrano-Ortiz, P., and Carrara, A.: Daily GPP estimates in Mediterranean ecosystems by combining remote sensing and meteorological data, ISPRS Journal of Photogrammetry and Remote Sensing, 102, 184-197, https://doi.org/10.1016/j.isprsjprs.2015.01.017, 2015.

Grove, A.T., Rackham, O. The Nature of Mediterranean Europe. An Ecological History. Yale University Press, New Haven and London, 2001.

Kowalski, A. S., Serrano-Ortiz, P., Janssens, I. A., Sánchez-Moral, S., Cuezva, S., Domingo, F., Were, A., and Alados-Arboledas, L.: Can flux tower research neglect geochemical CO2 exchange?, Agricultural and Forest Meteorology, 148, 1045-1054, 2008.

Lasslop, G., Reichstein, M., Papale, D., Richardson, A. D., Arneth, A., Barr, A., Stoy, P., and Wohlfahrt, G.: Separation of net ecosystem exchange into assimilation and respiration using a light response curve approach: critical issues and global evaluation, Global Change Biology, 16, 187-208, 10.1111/j.1365-2486.2009.02041.x, 2010.

López-Ballesteros, A., Serrano-Ortiz, P., Kowalski, A. S., Sánchez-Cañete, E. P., Scott, R. L., and Domingo, F.: Subterranean ventilation of allochthonous CO2 governs net CO2 exchange in a semiarid Mediterranean grassland, Agricultural and Forest Meteorology, 234–235, 115-126, http://dx.doi.org/10.1016/j.agrformet.2016.12.021, 2017.

Pérez-Priego, O., Serrano-Ortiz, P., Sánchez-Cañete, E. P., Domingo, F., and Kowalski, A. S.: Isolating the effect of subterranean ventilation on CO2 emissions from drylands to the atmosphere, Agricultural and Forest Meteorology, 180, 194-202, 2013.

Puigdefabregas, J., Mendizabal, T. Prospects of desertification impacts in western Europe. In: Marquina, A. (Ed.), Environmental Challenges in the Mediterranean 2000.2050. NATO Science Series. IV. Earth and environmental Sciences, p. 155, 2004.

Sanchez-Cañete, E. P., Serrano-Ortiz, P., Kowalski, A. S., Oyonarte, C., and Domingo, F.: Subterranean CO2 ventilation and its role in the net ecosystem carbon balance of a karstic shrubland, Geophysical Research Letters, 38, 2011.

Sánchez-Cañete, E. P., Kowalski, A. S., Serrano-Ortiz, P., Pérez-Priego, O., and Domingo, F.: Deep CO2 soil inhalation/exhalation induced by synoptic pressure changes and atmospheric tides in a carbonated semiarid steppe, Biogeosciences, 10, 6591-6600, 2013.

Serrano-Ortiz, P., Domingo, F., Cazorla, A., Were, A., Cuezva, S., Villagarcía, L., Alados-Arboledas, L., and Kowalski, A. S.: Interannual CO2 exchange of a sparse Mediterranean shrubland on a carbonaceous substrate, Journal of Geophysical Research G: Biogeosciences, 114, 2009.

---

## Editor Decision (ED1)

The authors of the manuscript (bg-2017-77) addressed the reviewers' issues clearly and answered questions reasonably. The manuscript has been largely improved. However, I still have some minor comments as listed below.

**Abstract:**

1. L20 on page 1, compared

2. L23 on page 1, south-east instead of 'SE'

3. L24 on page 1, I would like to change the sentence "Results show a striking difference in the annual C balances with an average release of 196 ± 40 and -23 ± 20 g C m-2 yr-1 for the "degraded" and "natural" sites, respectively." into "Results show a striking difference in the annual C balances with an average value of 196 ± 40 and -23 ± 20 g C m-2 yr-1 for the "degraded" and "natural" sites (minus represents a carbon sink), respectively."

4. The statement "we also tested differences in all monitored meteorological and soil variables and found it most relevant that CO2 at 1.50 m belowground was around 1000 ppm higher in the "degraded" site." is not clear to me.

5. In abstract, please weaken the wording of "subterranean ventilation of this vadose zone CO2, ……, largely drives the differences in C dynamics ……", since GPP at natural site was nine times that at degraded site. Therefore, the C dynamics was also driven biologically.

6. Delete the last sentence in the abstract, because it does not make a sense.

**Introduction:**

1. L20 on page 3, delete '(biological processes)' since ecosystem respiration also include non-biological process.

2. Move last sentence in Introduction to the Material and Methods section.

**Materials and Methods:**

1. Delete the sentence "Furthermore, over the six years of measurements at both sites, data gaps due to low-turbulence conditions, instrument malfunction and theft were unavoidable and not randomly distributed, as noted by Ma et al. (2016)."

2. L27 on page 5, rewording the sentence "In order to test the validity of both EC stations,'

3. L30-31 on page 5, rewording the sentence " Storage term in the soil heat flux….. the vegetation (~50 cm)"

4. L2 on page 6, gross primary production (GPP)

5. L7 on page 6, why you select the data base on "(ii) 8 < daily averaged bowen ratio < 10, and (iii) daily soil water content (in bare soil) < 10th percentile (in AMO) and < 20th percentile (in BB)."

**Results:**

1. L14 on page 8, with annual net C release of 240 ±8 in AMO and net C uptake of 38 ±10 g C m-2 in BB.  Please change the wording regarding net ecosystem production (NEP) throughout text.  Positive value of NEP represents C source or net C release, negative value represents C source or net C uptake.

2. L28 on page 8, delete the word 'fluxes' here.

3. L9 on page 10, "Firstly, GPP was always higher at BB compared to AMO (Fig. 7c)." .  Not really for the year 11/12 in Fig. 7c?

**Discussion**

1. L8 on page 12,  influencing GPP, $R_{eco}$, ….

2. L26-28 on page 12, I think also higher root respiration for natural site.

**Conclusions**

1. L9-10 on page 14, again change the statement into "annual average net C release of 196 ± 40 for the natural site and net C uptake of 23 ± 20 g C m-2 yr-1 for the "degraded" site were observed"

---

## Author Response (AR2)

**Reply to the editor's comments**

The authors of the manuscript (bg-2017-77) addressed the reviewers' issues clearly and answered questions reasonably. The manuscript has been largely improved. However, I still have some minor comments as listed below.

*We thank the editor for noticing the great effort we did during the revision procedure and for his minor remarks to improve this manuscript. Replies are shown below.*

**Abstract:**

1. L20 on page 1, compared

*We have followed your suggestion.*

2. L23 on page 1, south-east instead of 'SE'

*Done.*

3. L24 on page 1, I would like to change the sentence "Results show a striking difference in the annual C balances with an average release of $196 \pm 40$ and $-23 \pm 20$ g C m-2 yr-1 for the "degraded" and "natural" sites, respectively." into "Results show a striking difference in the annual C balances with an average value of $196 \pm 40$ and $-23 \pm 20$ g C m-2 yr-1 for the "degraded" and "natural" sites (minus represents a carbon sink), respectively."

*We agree with your suggestion and we have rewritten the sentence as follows: "Results show a striking difference in the annual C balances with an average net $CO_2$ exchange of $196 \pm 40$ (C release) and $-23 \pm 20$ g C $m^{-2}$ $yr^{-1}$ (C fixation) for the "degraded" and "natural" sites, respectively."*

4. The statement "we also tested differences in all monitored meteorological and soil variables and found it most relevant that CO2 at 1.50 m belowground was around 1000 ppm higher in the "degraded" site." is not clear to me.

*We have rewritten this sentence as follows: "We also tested differences in all monitored meteorological and soil variables and $CO_2$ at 1.50 m belowground was the variable showing the greatest inter-site difference, with~1000 ppm higher at the "degraded" site."*

5. In abstract, please weaken the wording of "subterranean ventilation of this vadose zone CO2, ……, largely drives the differences in C dynamics ……", since GPP at natural site

was nine times that at degraded site. Therefore, the C dynamics was also driven biologically.

*We have rewritten this sentence as follows: "Thus, we believe that subterranean ventilation of this vadose zone $CO_2$, previously observed at both sites, partly drives the differences in C dynamics between them, especially during the dry season maybe due to enhanced subsoil-atmosphere interconnectivity at the "degraded" site."*

6. Delete the last sentence in the abstract, because it does not make a sense.

*Done.*

**Introduction:**

1. L20 on page 3, delete '(biological processes)' since ecosystem respiration also include non-biological process.

*Done.*

2. Move last sentence in Introduction to the Material and Methods section.

*Done.*

**Materials and Methods:**

1. Delete the sentence "Furthermore, over the six years of measurements at both sites, data gaps due to low-turbulence conditions, instrument malfunction and theft were unavoidable and not randomly distributed, as noted by Ma et al. (2016)."

*Done.*

2. L27 on page 5, rewording the sentence "In order to test the validity of both EC stations,'

*We have rewritten this sentence as follows: "In order to test the validity of both EC stations, we assessed the energy balance closure (Moncrieff et al., 1997) by computing the linear regression of half-hourly turbulent energy fluxes (sensible and latent heat fluxes; H+LE; W $m^{-2}$) against available energy (net radiation less the soil heat flux; $R_n$-G; W $m^{-2}$) using the whole six-years database."*

3. L30-31 on page 5, rewording the sentence " Storage term in the soil heat flux….. the vegetation (~50 cm)"

*We have added a comma in this sentence as follows: "The storage term in the soil heat flux was included in the estimates, while in the cases of sensible and latent heat fluxes, this term was negligible given the short height of the vegetation (~50 cm)."*

4. L2 on page 6, gross primary production (GPP)

*Done.*

5. L7 on page 6, why you select the data base on "(ii) 8 < daily averaged bowen ratio < 10, and (iii) daily soil water content (in bare soil) < 10th percentile (in AMO) and < 20th percentile (in BB)."

*As stated before this sentence, we have used previous results included in two publications where data from both sites were analyzed. To make it clear, we have modified the following sentence as follows (lines 6-7, page 6): "we firstly modelled the ventilative $CO_2$ efflux by adapting the approach proposed by Pérez-Priego et al. (2013) using the results of previous studies at both sites (López-Ballesteros et al., 2016; 2017)."*

**Results:**

1. L14 on page 8, with annual net C release of 240 ±8 in AMO and net C uptake of 38 ±10 g C m-2 in BB. Please change the wording regarding net ecosystem production (NEP) throughout text. Positive value of NEP represents C source or net C release, negative value represents C source or net C uptake.

*We disagree with the editor on this point. On one hand, NEP means Net Ecosystem Productivity, therefore negative values correspond to carbon emission and positive ones to carbon fixation. On the other hand, NEE, which means Net Ecosystem Exchange, represents the same concept but from the atmospheric point of view, where negative and positive $CO_2$ fluxes correspond to carbon fixation and emission/release, respectively. However, we avoided the widely used term Net Ecosystem Exchange (NEE) to refer to the net $CO_2$ exchange we measured because some of the $CO_2$ released in Amoladeras is probably not exclusively local (López-Ballesteros et al. 2017) and because part of the vadose zone is actually beyond the ecosystem conceptual boundaries (Chapin et al., 2006).*

*Nonetheless, we have rewritten the sentence to clarify the results as follows: "The year with the largest difference between sites was 2010/2011, with annual C release of 240 ±8 and annual C uptake of -38 ±10 g C m$^{-2}$ in AMO and BB, respectively (Fig. 3b)."*

*Chapin, F.S., Woodwell, G.M., Randerson, J.T., 2006. Reconciling carbon-cycle concepts, terminology, and methods. Ecosystems 9, 1041–1050.*

*López-Ballesteros, A., Serrano-Ortiz, P., Kowalski, A. S., Sánchez-Cañete, E. P., Scott, R. L., and Domingo, F.: Subterranean ventilation of allochthonous CO2 governs net $CO_2$ exchange in a semiarid Mediterranean grassland, Agricultural and Forest Meteorology, 234–235, 115-126, http://dx.doi.org/10.1016/j.agrformet.2016.12.021, 2017.*

2. L28 on page 8, delete the word 'fluxes' here.

5    *Done.*

3. L9 on page 10, "Firstly, GPP was always higher at BB compared to AMO (Fig. 7c)." . Not really for the year 11/12 in Fig. 7c?

*This corresponds to a lack of GPP data in Balsa Blanca.*

**Discussion**

10    1. L8 on page 12, influencing GPP, Reco, …

*Done.*

2. L26-28 on page 12, I think also higher root respiration for natural site.

*We agree and we have rewritten this sentence as follows: "On one hand, $CO_{2,\ 0.05m}$ was generally higher at the "natural" site given its lower degradation level, which probably*
15    *promotes higher root respiration and microbial activity supported by higher vegetation density and soil fertility (Table 1) especially during spring (Fig. 8), as pointed also by Oyonarte et al. (2012)."*

**Conclusions**

1. L9-10 on page 14, again change the statement into "annual average net C release of 196 ±
20    40 for the natural site and net C uptake of 23 ± 20 g C m-2 yr-1 for the "degraded" site were observed"

*We have changed the sentence as noted in the reply above (abstract).*

[revised manuscript text omitted]
}$ | 0 | -0.28 | -0.28 | -0.49 | -0.64 | -0.74 | 0.00 | -0.13 | -0.28 | -0.43 | -0.57 | -0.48 |